# Top-down identification of keystone taxa in the microbiome

**Guy Amit** [1,2] **& Amir Bashan** [1] ✉

Keystone taxa in ecological communities are native taxa that play an especially important role in the stability of their ecosystem. However, we still lack an effective framework for identifying these taxa from the available high-throughput sequencing without the notoriously difficult step of reconstructing the detailed network of inter-specific interactions. In addition, while most microbial interaction models assume pair-wise relationships, it is yet unclear whether pair-wise interactions dominate the system, or whether higher-order interactions are relevant. Here we propose a top-down identification framework, which detects keystones by their total influence on the rest of the taxa. Our method does not assume a priori knowledge of pairwise interactions or any specific underlying dynamics and is appropriate to both perturbation experiments and metagenomic cross-sectional surveys. When applied to real high-throughput sequencing of the human gastrointestinal microbiome, we detect a set of candidate keystones and find that they are often part of a keystone module – multiple candidate keystone species with correlated occurrence. The keystone analysis of single-time-point cross-sectional data is also later verified by the evaluation of two-time-points longitudinal sampling. Our framework represents a necessary advancement towards the reliable identification of these key players of complex, real-world microbial communities.

The concept of keystone taxa (species), which was first introduced in 1969 by Paine[1,2], generally refers to native species that play an especially important role in the stability of their ecosystem. Since then, identifying keystones has become an elemental component in analyzing ecosystems in order to understand their vulnerabilities and maintain sustainability[3,4]. Keystone species could also be potentially used in modulation experiments, as main drivers of an existing system to an alternative, more desirable, steady state[5,6]. An exact ecological definition of keystone species has been a subject of a long-standing debate[3,7] and has been continually developed over the years[8].

One well-accepted definition for keystone species was given by Power et al.[9]. There, the authors introduced the term 'community importance', which evaluates the effect of a species on traits of the ecosystem, such as productivity, nutrient cycling, species richness, or the abundance of one or more functional groups of species or of dominant species. Community importance was defined in one of two ways. The first was the abundance-to-trait relationship, which tests how much the relative abundance of a species affects the trait in question. The second was the presence-to-trait relationship, which tests how the removal of a species entirely from the system affects the trait. A species with unusually large community importance, either by its abundance-to-trait relationship or presence-to-trait, is considered to be a keystone species. We refer to these definitions as 'abundance-impact' and 'presence-impact', respectively (see Table 1).

Both ways define the 'keystoneness' of a species through an ideal experimental setup in which its abundance, or presence, is controlled and altered directly. Such experiments are indeed the only direct way of identifying a species as a keystone, since they can capture all the

[1]Department of Physics, Bar-Ilan University, Ramat-Gan 590002, Israel. [2]Department of Natural Sciences, The Open University of Israel, Raanana 4353701, Israel. ✉e-mail: amir.bashan@biu.ac.il

**Table 1 | Glossary of often-used terms throughout the manuscript**

| Term | Definition |
|---|---|
| Simulations of ecological dynamics | |
| GLV | Generalized Lotka-Volterra. |
| Interaction network | The intrinsic dynamical rules governing the interactions between the species. |
| Interaction-strength-based keystones | Artificial keystones generated by modifying a species interactions strengths. |
| Structure-based keystones | Artificial keystones are generated by the natural structure of the interaction network. |
| Keystoneness definition | |
| Community Importance | The effect of a single species on traits of the ecosystem. |
| Presence-impact | The effect of a single species on the rest of the species in the community, as measured by addition/removal experiments (also termed 'presence-to-community-impact'). |
| Abundance-impact | The effect of a single species on the rest of the species in the community is hypothetically measured by abundance alteration experiments (also termed 'abundance-to-community-impact'). |
| Keystone candidates identification | |
| Co-occurrence network | The empirical network of correlations between the species was reconstructed from cross-sectional samples. |
| EPI | Empirical presence-abundance interrelation; the estimated level of presence-impact of a species based on cross-sectional data. |
| Longitudinal EPI | The estimated level of presence-impact of a species based on longitudinal data. |

complex, and unexpected consequences arising from non-linear and often indirect interactions. It should be noted that, in contrast to presence-impact experiments, abundance-impact perturbation might be practically impossible in real-world experiments[9].

Recent data-driven research of microbial communities provides new opportunities for finding keystone species, but also poses new challenges[5]. One major challenge stems from the fact that studies of natural microbial communities, such as environmental or human-associated microbiomes, commonly do not involve controlled perturbation experiments, due to both technical and/or ethical reasons. Instead, they are usually studied through large-scale cross-sectional metagenomic surveys. These surveys are typically rich in data, composed of hundreds of metagenomic samples which contain thousands of species[10,11]. Without perturbation experiments, the research is focused on identifying candidate keystones, by estimating the species impact from the cross-sectional data alone[12].

Traditional identification methods of candidate keystones rely on evaluating their centrality in a mediation network model, such as co-occurrence networks or inferred models of the underlying dynamics, e.g., parameterization of Generalized Lotka-Volterra (GLV) or consumer-resources models[13–17]. This approach has several fundamental drawbacks[18]. For example, complete reconstruction of the ecological network of $N$ species from cross-sectional data in a bottom-up fashion is very challenging, since the number of available samples is typically much smaller compared to the number of possible pair-wise interactions, $N^2$. In addition, conventional correlation analysis is subject to spurious correlations due to the compositionality of relative abundances in genomic survey data[19,20]. Furthermore, mediation network models are based on the assumption that interspecific interactions are pair-wise with a specific functional form. Another drawback, which is more pertinent, is that the commonly used interpretation of keystone species focuses on their presence-impact (i.e., 'how will the system react to a removal of the species?'). This interpretation coincides well with the currently available manipulation techniques of microbial communities, which mainly include controlling the presence of species, e.g., by removing or adding species using antibiotics, probiotics, or fecal microbiota transplant, and seldom by direct control of their abundance. In contrast, these network-based models measure instead the abundance impact of the species as they analyze how the abundance of each species is related to the other species.

Therefore, there is a conceptual and practical gap between the species with the high presence-impact we aim to detect and the abundance impact measured by traditional network models.

Here, to overcome these difficulties, we introduce a unified framework for identifying keystone species that can be applied consistently to both simulated perturbation experiments and cross-sectional data. A keystone's presence-impact is defined by how much its presence or absence affects the abundance profile of the rest of the species in a top-down manner. The effect of the species is measured without calculating the pair-wise correlation network and does not even assume that the ecological dynamics are governed by pair-wise interactions. This network-free approach to measuring the presence-impact of species avoids the above-mentioned pitfalls of network reconstruction.

## Results

As mentioned above, the presence-impact of a species can only be definitively determined using removal/addition perturbation experiments, while analysis of cross-sectional data can indirectly estimate the species' impact. Following the methodology presented in refs. 9,21, we introduce a presence-impact measure that can be applied to both perturbation experiments and cross-sectional data. The presence-impact $I^i$ of species $i$ is determined by the change of the abundance profile of all other species in its local community associated with reversing its presence state, i.e., removing the species if it was present, or introducing it into the system if it was absent (See Fig. 1a and Methods). Likewise, if a species has a strong presence-impact, as measured using perturbation experiments, its presence/absence pattern in naturally assembled communities is also expected to be associated with community-wise differences in the abundance profiles of the other species. This empirical association, which we term Empirical Presence-abundance Interrelation (EPI), can be detected from cross-sectional data to identify candidate keystone species. Specifically, for a given species $i$, we propose three different definitions of its EPI: $D_1^i$ and $D_2^i$ which are based on the distances between the relative abundance profiles and $Q^i$ which is based on the modularity concept from network science (see Fig. 1b–d and Methods). The main definitions used in this manuscript are summarized in Table 1 and a list of the mathematical symbols is presented in Supplementary Table 1. Note the distinction here between candidate keystones and actual keystone taxa. Since the EPI is measured only on cross-sectional data, it can not distinguish between correlation and causation effects. In other words, a taxon can have a large EPI value if it affects the other taxa disproportionately, or if it is affected disproportionately by other species. To disentangle this correlation from causation effects, perturbation experiments are required, where the actual presence-impact can be directly measured.

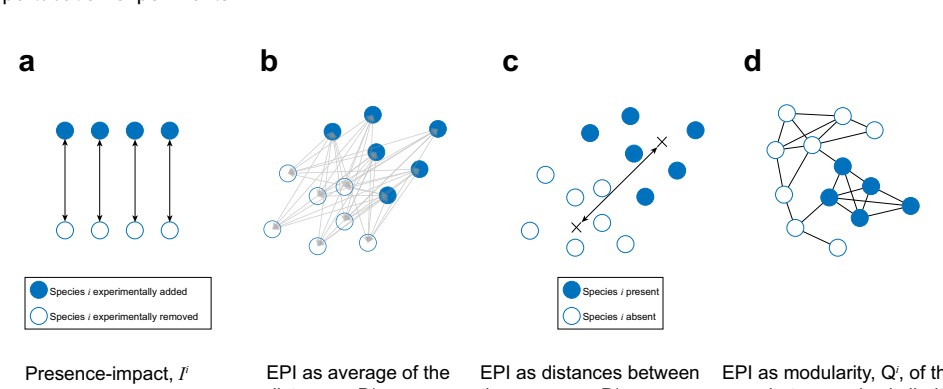

**Fig. 1 | Schematic illustration of presence-impact calculation. a** Presence-impact, $I^i$, as measured by perturbations experiments, is defined by the distance between the abundance profiles of the same community before and after species $i$ is added/removed. **b, c** Two distance-based measures of estimating the EPI from cross-sectional data. The first, $D_1^i$, is by measuring the mean distance between all pairs of samples with and without the species (represented by the gray arrows). The second, $D_2^i$, is by measuring the distance between the means of the groups, represented by the black arrow between the two crosses. **d** EPI using the modularity measure, $Q^i$. A sample-to-sample similarity network is constructed and its modularity is calculated based on the two groups of samples, those where species $i$ is present and those where it is absent. High modularity of the network indicates a natural separation between the samples associated with the presence-absence of the species. The distances between the samples in all cases are measured with respect to the re-normalized abundance of all other species (see Methods).

The EPI proposes which species of interest will be leading candidates for further perturbation experiments.

Here we use numerical models of population dynamics with artificial keystones that allow us to simulate and study the relations between three aspects: the underlying interactions, the presence-impact from numerical experiments, and the EPI from simulated cross-sectional data. Later on, we apply the EPI to real cross-sectional high-throughput sequencing to demonstrate it in naturally complex communities.

**Presence-impact in simulations**

We start with a simple demonstration of the presence-impact definition, $I^i$, on simulated GLV perturbation experiments with designated strength-based and structure-based keystones (see Methods). As shown in Fig 2, the presence-impacts of the designated strength-based and structure-based keystones are significantly larger than those of the non-keystone species (see Methods for the statistical tests used). This demonstrates that the underlying dynamics of the ecological community are manifested in the presence-impact as measured from the resulted abundance profile. Note that we define the presence-impact through the abundance change of the other species. This has three main advantages, compared with a previous definition which considers the number of extinct species[22]: (i) The impact of a keystone species considers not only its negative influence on other species that may lead to their extinction but also the positive influence that increase their abundance. (ii) There is no need to set an extinction threshold when calculating the impact from perturbation experiments. (iii) The measured impact is found to be closely related to the underlying dynamical structure (see Supplementary Fig. 3).

In order to study the relation between the underlying dynamics and cross-sectional data we first simulate cohorts of cross-sectional data with a single designated strength-based keystone and measure the EPI of all the species (Fig. 3). In this ideal case, all three above-mentioned EPI measures ($D_1$, $D_2$, and $Q$) successfully mark the designated species as a clear keystone candidate, as its EPI values are significantly larger compared with the rest of the species. This can be seen in the PCoA space where the samples are distinctly divided into two groups, depending on whether the keystone species is present or absent (Fig. 3b, e), while the high modularity of the keystone species is

evident by the fact that there are only a few samples of different groups that are connected with an edge (Fig. 3h).

Next, we conduct systematic experiments in order to test the statistical relationship between the presence-impact, as measured from perturbation experiments, and the EPI, as measured from cross-sectional data (Supplementary Fig. 3). We define ecological dynamics with either strength-based or structure-based keystones, simulate cohorts of $M$ samples with $N$ species, and perform perturbation experiments for each of the species. We then test the correlation between the three EPI measures ($D_1^i$, $D_2^i$ and $Q^i$) and the presence-impact $I^i$, for all $i = 1, ...N$. To ensure a wide range of impact values in the case of the strength-based keystones, the boosting parameter, $K^i$, was taken from a log-normal distribution with parameters $\mu = 0.5$ and $\sigma = 1$. The range of values of the presence-impact in the structure-based case is due to the natural scale-free degree distribution of the network. As shown in Supplementary Fig. 3, significant correlations are evident between each of the EPI measures and the experimental presence-impact $I$. Notably, the modularity measure $Q$ had the largest significant correlation amongst the EPI measures.

In Supplementary Fig. 6 the abundance value. In other words, while a PCA plot might show a large difference between samples based on large variability in the relative abundance values of species, even when interactions are not present, the EPI measure removes the test species from the data before calculating its impact is less sensitive to such variations.

In Supplementary Fig. 7 we show how the $D_1$ EPI measure is used to identify keystone species in the Stable Marriage model of ecological dynamics[23]. In this model, the species abundance is determined by preference lists, not an underlying interaction network, so detecting species using a reconstruction of the correlation network makes little sense. When the preference lists are correlated, the system is frustrated and keystone species with higher-than-average EPI values are prominent.

**Presence-impact in real microbial communities**

We analyze real metagenomic cross-sectional data of the gastro-intestinal tract from the Human Microbiome Project (HMP)[24,25] (see Methods) and apply our EPI measures to identify keystone candidates. Estimation of species abundances through metagenomic sequencing

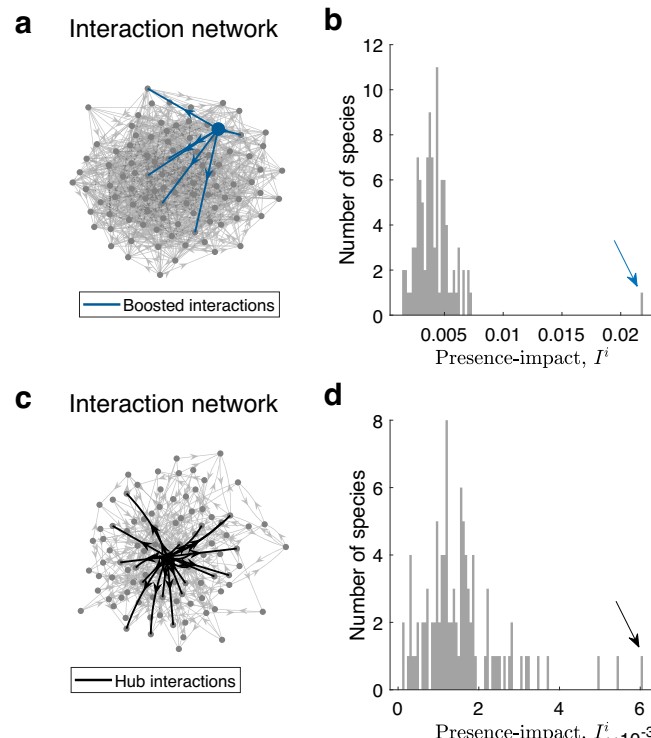

**Fig. 2 | Detection of a keystone species in simulated perturbation experiments.**
**a** Interaction-strength-based keystone in an Erdős-Renyi network. The network represents the interaction matrix $A$ of $N = 100$ species. The average degree $\langle d \rangle$ of the nodes (species) is 10. The directed blue edges represent the out-going interactions of a designated keystone species $j$, which were boosted with a constant $K^j = 10$. Based on this network, perturbation experiments were performed for each species as detailed in Methods. **b** Presence-impact of each species of the network in **a**. The arrow marks the impact of the designated keystone species, which is significantly above the rest of the species ($p = 8.242 \times 10^{-18}$ using a one-tailed $z$-test). **c** Structure-based keystone created using Barábasi-Albert model, with parameters $m_0 = 3$ and $d = 0.1$ (see Methods). Here the strengths of the interactions remain untempered with. However, there are natural hubs, i.e., species with large number of interactions. The largest hub and its out-going interactions are marked in solid black. **d** Presence-impact of each species of the network in **c**. The arrow marks the impact of the hub species, which is significantly larger than the rest of the species ($p = 1.004 \times 10^{-6}$ using a one-tailed $z$ test). The few remaining species with high impact are also natural hubs, albeit smaller than the main one. For both cases, distances between the reduced abundance profiles, $\langle S_k^i, S_k^{i*} \rangle$, were measured using the Bray–Curtis dissimilarity measure.

surveys is susceptible to significant uncertainties due to many factors including experimental errors and sampling noise. This also affects the presence/absence pattern of the species. For example, it has been recently shown[26] that the species observed relative frequencies (the percent of samples where the species were detected) is determined by the mean and variance of their abundances, together with the sampling depth. Furthermore, the presence/absence of data is affected by the specific experimental and computational pipelines used. Thus, when analyzing real high-throughput sequencing, the 'absence' of a species should be interpreted as being below the detection limit, whereas an observed species is more confidently defined as 'present'. There are, however, arguments in favor of considering presence/absence data in high-throughput sequencing[27], specifically with regards to the problem of compositionality, however the opinions on that subject are still divided. The EPI measures use a combination of both the presence/absence of the species in question and the relative abundance data of the other species. We choose to focus on the presence/absence of the species in question, mainly due to two

methodological reasons: (i) The original and commonly used ecological definition of keystone species relates to its presence/absence. (ii) Currently available methods for manipulating microbial communities include tools that can introduce new taxa (probiotics and FMT) or remove taxa (narrow spectrum antibiotics), while directly controlling the abundance of specific species is currently not feasible.

Figure 4 presents the EPI values, of all three measures, for $N = 1000$ top-abundant species (OTUs). The figure shows the existence of candidate keystones in real data from stool samples, as exemplified by EPI values larger than two standard deviations compared with the rest of the species (marked by the shaded gray area). Note that keystone candidates are identified only by the relative values of their EPI compared with the cohort of the species, not by the absolute value. For each of the measures, the separation between the samples associated with the presence/absence of the taxa with the highest EPI value is clear in the PCoA plots and network model (Fig. 4b, e, h) in marked contrast with a random, non-keystone, species (Fig. 4c, f, i). These three taxa detected by the different EPI measures are defined as distinct OTUs, nevertheless, their taxonomic classification is identical, i.e., all three OTUs were classified down to the *Bacteroides* genus level (but have no classification at the species level). In addition, the EPI values calculated for all OTUs by the different measures are significantly correlated as shown in Supplementary Fig. 4.

Systematic analyses of additional 12 cohorts of microbial samples from different body sites are shown in Supplementary Figs. 12–24. In most cases, we see that the distribution of EPI values has a dominant mean peak with a small number of species with considerably higher EPI values. In addition, the presence/absence of species with the top EPI value is associated with the separation of the samples in the PCoA plots and the sample-to-sample networks. Exceptional is the vaginal microbiome (Supplementary Figs. 22–24), which is characterized by obvious clusters, or state types[28] (as also seen in the PCoA analysis). In these cases, the presence/absence patterns of many species are associated with the clusters, so no specific species stand out in the EPI distributions. In the Discussion section, we further discuss the relationship between the EPI method and dimension reduction techniques, such as PCA or PCoA.

As previously stated, direct validation of the EPI requires perturbation experiments on a large number of taxa. In Supplementary Fig. 8 we show additional analysis of keystone detection in the leaves microbiome of the plant model *Arabidopsis*. The "effect size" from perturbation experiments for a small number of selected species of one data set (see ref. 29 for further details) is compared with the EPI values calculated from cross-sectional data of a different experiment[30]. We find a significant correlation between the two values ($r = 0.7926$ with $p = 0.00367$), suggesting that natural variability across samples is shaped by ecological interactions that can be observed by controlled perturbation experiments. In addition, even though the sequencing depth is a major contributor to biases in the presence/absence of data[26], in Supplementary Fig. 9 we show that the ranking of the taxa according to the EPI values, of the top keystone candidate, is relatively stable to the choice of sequencing depth in both simulated and real data from the human gut.

**Longitudinal EPI**

When calculating the EPI from cross-sectional data, two possible issues may arise. First, the empirical association observed for a given species may be due to confounding factors attributed to inter-personal heterogeneity in the analyzed population, such as genetics, life-style, or nutritional variability. Second, while our original definition of the presence-impact of a species considers the magnitude of the transition of its microbial community following perturbation experiments, the EPI is measured across different subjects for which a direct transition may not be induced by that species. To address these issues, we analyze the presence-impact using an alternative

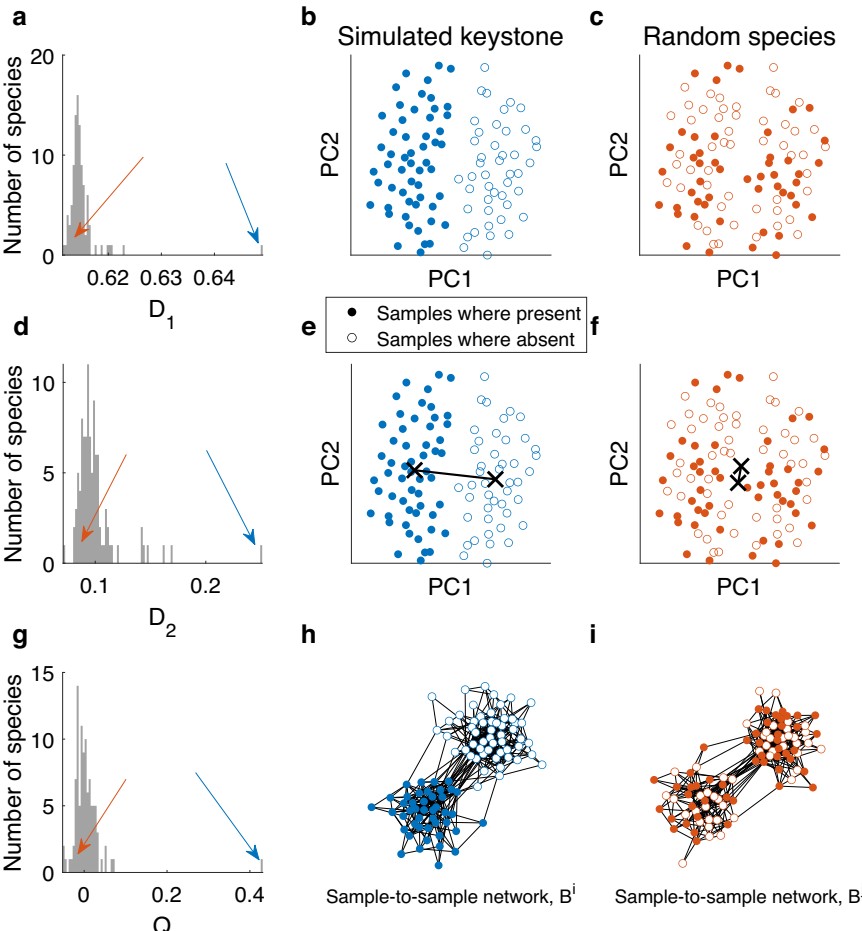

**Fig. 3 | Demonstration of the three EPI measures calculated on simulated cross-sectional data.** Cross-sectional samples were simulated with a single designated interaction-strength-based keystone (see Methods section). **a** Distribution of the EPI $D_1$ values of all the species. The solid blue arrow points to the $D_1^i$ value of the keystone species, $i$, and the red arrow points to the $D_1^j$ value of a random species, $j$. **b** PCoA visualization of the abundance profiles $S_k^i$ colored by the presence/absence of the keystone species. The clear separation between the two groups illustrates the high EPI value of the keystone species. **c** Similar to **b** for the random species $j$. Here there is no separation of the samples. **d**–**f** Similar to **a**–**c** for the EPI $D_2$. The black crosses in **b**–**f** mark the mean of the groups. **g** Similar to **a** for the modularity EPI measure, $Q$. **h** The sample-to-sample correlation network, $B^i$, associated with the keystone species $i$. Filled (empty) nodes represent samples where the species is present (absent). The natural separation between the nodes into two groups indicates the large modularity value $Q^i$. **i**, Similar to **h** for a random species $j$. The lack of separation between the groups indicates the low modularity value $Q^j$ of the random species. Samples were generated using GLV simulations on an Erdős-Renyi network with $N = 100$ nodes (species). The average degree of each node is 50. The internal growth rate of each species $i$, $r_i$ was set to unity. The boosting parameter was set to $K = 10$. The samples were normalized to 1, and the distance metric used was Bray–Curtis. The network threshold parameter for the calculation of the modularity was $p_Q = 0.1$.

approach from two-time-points longitudinal data and compare it with the results from cross-sectional data (see Fig. 5a). The longitudinal EPI, $L_k^i$, measures the dissimilarity between two samples from the same body site collected from the same subject, $k$, with a time interval, where the presence state of species $i$ is different between the two samples. This is then averaged across all subjects to get a single value $L^i$ (see Methods). Figure 5b shows the relationship between the empirical and longitudinal EPI. The two measures are significantly correlated (Pearson coefficient of $r = 0.38$ with $p < 10^{-17}$). Specifically, there is a significant agreement between the sets of candidate keystones from the two measures, with 11 shared candidates ($p < 10^{-11}$ using the Fisher test). Furthermore, after a shuffling process that preserves both the relative frequencies of the species and the number of observed species in each sample, the presence-impact shows no agreement between the EPI and the longitudinal EPI (see Supplementary Fig. 5). The general agreement between the EPI and the longitudinal EPI, which is less prone to the above-mentioned issues, further supports that the presence-impact of a species can be captured from the analysis of cross-sectional data.

The methodology of comparing the cross-sectional EPI and the longitudinal presence-impact has two limitations. First, the calculation of the longitudinal presence-impact for a particular microbial species is possible only using a subset of the population for which the presence attribute of that species has changed between the first and second measurements. Second, although the host-related environmental factors are assumed to be more stable for longitudinal data, it is possible that some confounding factors are related to both across-hosts and within-host variability, leading to spurious correlations between the two calculated measures. This effect can be completely ruled out only by direct perturbation experiments. An example of how environmental confounding factors may affect keystone candidates detection is presented in Supplementary Fig. 25.

**Keystone modules**

An interesting question is how the presence-impact of the species is related to the co-occurrence relationships of different species. To investigate this, we construct a co-occurrence network where the nodes represent the different species, and the links are based on the

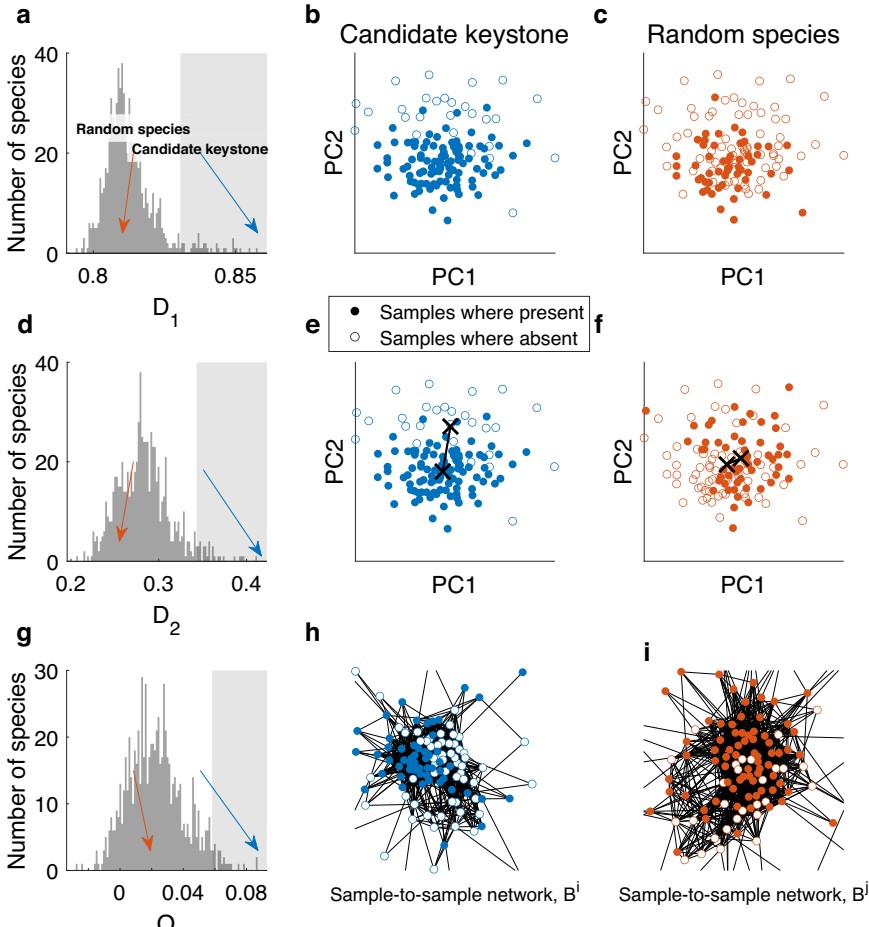

**Fig. 4 | EPI of real high-throughput sequencing from the gut microbiome.**
**a** Distribution of the EPI $D_1$ values of all $N = 1000$ top-abundant species. The gray area marks the EPI values greater than two standard deviations from the mean. Blue and red arrows mark the EPI values of a candidate keystone, $i$, and a random species, $j$, respectively. **b** PCoA visualization of keystone-associated abundance profiles $S_k^i$. Filled dots represent samples where the species is present, empty circles represent samples where the species is absent. The samples are naturally separated by the absence/presence of the keystone species into two types. **c** Similar to **b** for the random species $j$. Here there is no visible separation of the samples into types. **d**–**f** Similar to **a**–**c** for the EPI $D_2$. The black crosses mark the mean of the groups.

**g** Similar to **a** for the modularity EPI measure, $Q$. **h** The sample-to-sample correlation network, $B^i$, associated with the keystone candidate $i$. Filled (empty) nodes represent samples where the species is present (absent). The natural separation between the nodes into two groups indicates the large modularity value $Q^i$. **i** Similar to **h** for a random species $j$. The lack of separation between the groups indicates the low modularity value $Q^j$ of the random species. The three EPI distributions clearly diverge from a normal distribution and are rather right-skewed (the percentages of species with EPI larger than two standard deviations from the mean are 4.6%, 4.1%, and 3.1%, for $D_1$, $D_2$, and $Q$, respectively, compared with the 2.2% expected in the case of normal distribution).

presence/absence relationship between them, calculated as the Normalized Mutual Information[31]. Figure 6 shows details from the networks, where each node is colored by its EPI value, calculated independently from the network itself. We find that many of the observed candidates are part of what we term a 'keystone module', a set of species with highly correlated presence-absence patterns, which together form a group that has a large presence-impact on the rest of the species. The black rectangles in Fig. 6 highlight examples of such groups of species with high mutual information between them, and generally large EPI values.

To test this effect, we compare the EPI values of species to their neighboring species in the mutual information co-occurrence network. As shown in Fig. 6d–f, the EPI values of neighboring species are significantly correlated. Such correlation is not observed in a null model, where the EPI values were randomly reshuffled between the species. This means that species have a tendency to share high EPI values with neighboring species, supporting the existence of such keystone modules. A reasonable hypothesis is that one or few of the species in the module is indeed a keystone species, and the rest are 'satellite species', which themselves do not have an unusually large

presence-impact, but are strongly connected to the keystone species. In Supplementary Fig. 10, we show similar results obtained with the Jaccard similarity measure. Future work should be done on identifying those keystone modules systematically, and distinguishing between the keystone species and the satellite species.

## Discussion
Due to recent advances in metagenomic sequencing, researchers have been able to study and characterize microbial communities under countless conditions and scenarios. Many of those studies also include a keystone identification step, since they are considered as main drivers of the ecological dynamics of the community. It is therefore vital for us to define what keystone species is in exact terms, and also have our detection protocols coincide with our definition. In this work we show how to calculate the presence-impact from perturbation experiments, and propose three alternative definitions for EPI from cross-sectional data to detect keystone candidates, all in a top-down manner without any network reconstruction.

These definitions are not mere semantics. Characterizing precisely the presence-impact of a species is important for practical

reasons: Current real microbial perturbation experiments (on human subjects or other environments) mainly rely on either introducing new species to the community or eliminating them. Directly controlling the abundance of specific species in a community is, at present, not feasible. The information on the species' presence-impact can be directly applied when designing such perturbations. on the one hand, perturbations, such as probiotics, that include keystone taxa should be treated with caution since they may lead to dramatic changes in the microbial community. On the other hand, keystone taxa may be used as the main drivers when steering the microbiome to a desired alternative state.

An interesting question is how the presence-impact of a species is related to its abundance impact. Assuming that we have a well-accepted definition of the influence of a species on a system[32,33], we can think about two different types of how the influence of a species depends on its abundance. In one case, the influence of a species grows with its abundance, i.e., 'abundance-impact'. In another case, the influence of a species is mainly dependent on its presence, but remains constant for different abundances, i.e., 'presence-impact' (as schematically illustrated in Supplementary Fig. 11). For example, the influence of a species that consumes common resources is expected to be dependent upon its abundance. In another example, a pathogen may indirectly affect the community by triggering a response of the immune system in a threshold-like manner, while above the triggering threshold, the influence remains the same. The EPI measure for cross-sectional data captures both effects, while a complimentary working definition of an abundance impact is required to distinguish between them.

In some cases, the effect of a keystone species can be captured using dimensional reduction techniques, as demonstrated in Fig. 3. Yet, this approach has two significant drawbacks. First, when the samples represent relative abundances, large variations between the samples can be caused by the variance of a high-abundant species which does not necessarily interact with the other species either directly or indirectly (see Supplementary Fig. 6. In contrast, the EPI of a particular species is calculated over the re-normalized abundance profiles excluding that species (3), such that the spurious correlations due to the compositionality of the data are removed. Second, the aim of the PCA/PCoA methods is to preserve the global structure of the data and thus may miss the local effect of the presence/absence of a species[34]. By local effects, we mean that the presence/absence effect is measured between communities with similar compositions. Such local effects are conceptually analogous to perturbation experiments where

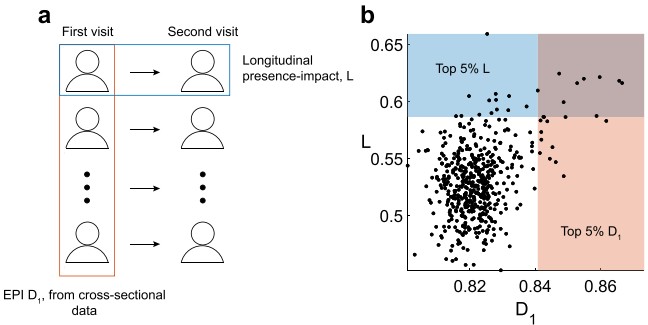

**Fig. 5 | Comparison between the longitudinal and empirical presence-abundance interrelation in the gut microbiome. a** The HMP data set includes 121 individuals for whom two stool samples have been collected with a time interval (between one and twelve months)[50]. We calculate the EPI $D_1$ values using the cross-sectional data from the first visits only, and compare them with the longitudinal EPI values, $L$, which are calculated by comparing the first and second samples of each subject individually (see Methods). **b** Each dot represents the $D_1$ and $L$ values of an individual species (After the filtering process, we are left with 509 species, see Methods). The values of the two measures are significantly correlated ($r = 0.38$, $p = 1.0082 \times 10^{-18}$ using Pearson correlation). The top 5% $D_1$ and $L$ values are marked by the shaded red and blue areas, respectively, which correspond to the candidate keystones of the two methods. Using Fisher test, we found a significant level of correspondence between the shared candidates of the two methods ($p = 5.1598 \times 10^{-11}$ with 11 shared candidates). Ten of the shared OTUs are of the genus *Bacteroides* and one is of *Leptotrichia*. All of them with unassigned species taxonomic level.

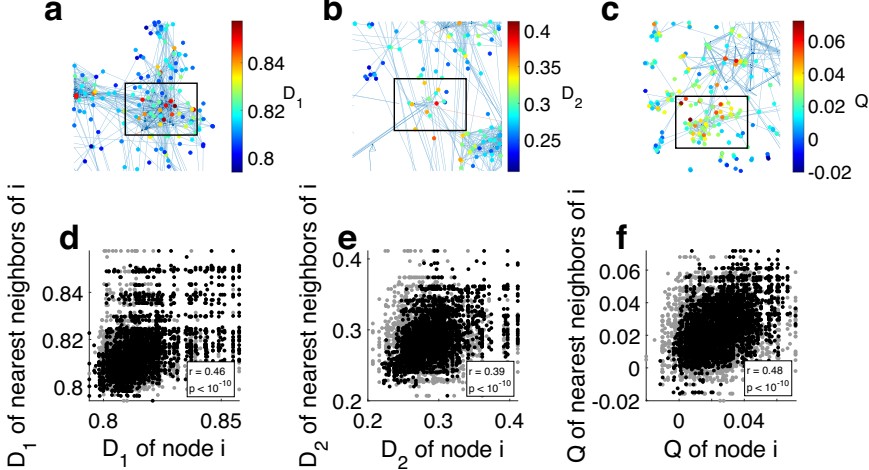

**Fig. 6 | Keystone modules in the presence-absence co-occurrence network.** Analysis was done on the top $N = 1000$ species in gut microbiome data. **a** Detail of the co-occurrence network of species based on the presence-absence data. Edges represent the top 25 percentile of normalized mutual information values calculated between all species pairs. The edges were colored according to the Pearson correlation, where blue (red) indicates positive (negative) correlations. Each node (species) in the network is colored by its EPI $D_1$ value. The black rectangle highlights an example of a typical group of highly correlated species which large EPI values. All the species in the rectangle are of the genus *Bacteroides*. **b** Similar to **a** for $D_2$. All the species in the rectangle are of the genus *Bacteroides*. **c** Similar to **a** for $Q$. All the species in the rectangle are of the genus *Faecalibacterium*. **d** We statistically study the relation between the network structure and the EPI values of the nodes by calculating the correlation between the EPI $D_1$ values of the nearest neighbor species (black dots). The gray dots represent the same values after randomly shuffling the EPI values among the species. Pearson correlation scores and associated $p$ values are presented in the figure. **e, f** Similar to **d** for $D_2$ and $Q$. From **d**–**f** the $p$ values are $p = 3.3241 \times 10^{-203}$, $p = 2.095 \times 10^{-178}$, $p = 1.4273 \times 10^{-127}$.

a single species is introduced to a community to measure how it affects that same community.

Focusing on the local effects is important due to two characteristics of real microbial communities. The species-species interactions may be context-dependent, i.e., the same species may have different effects in different microbial communities[35–38]. In addition, microbial interactions are inherently non-linear[39,40], and in fact, it has been recently shown that straightforward correlation analysis of pair-wise interactions from cross-sectional abundance information carries only limited information on the real underlying interactions[14,41]. The PCA/PCoA methods are not designed to account for context-dependent and non-linear interactions[34].

The EPI measures $D_1$ potentially circumvents this difficulty as it does not assume the samples are divided into two distinct groups, separated along a single dimension, but rather takes into account more complex effects, as schematically demonstrated in Fig. 1b, c. The $Q$ is designed to be even more sensitive to non-linear effects, as it compares only neighboring samples when evaluating the effect of the species, methodologically similar to the t-SNE method. When analyzing real-world data, it is important to utilize all the methods in our arsenal to ensure robust statistical conclusions, and the EPI measure helps us enforce the validity of keystone identification protocols.

When analyzing a given data set, the top-down approach, such as the EPI method, and the bottom-up approach, such as the reconstruction of a correlation network, provide two different perspectives on the same system. We believe that these approaches can complement each other. In one direction, information from the network can be reduced to its components, i.e., detecting important species by their centrality in the network, which can be compared with the EPI results. Moreover, top-down information can be incorporated to facilitate network analysis. An example of such a dual approach is the observation of keystone modules. It would be interesting to devise a way to systematically identify them from a given data. Later on, we hope to be able to distinguish between real keystones in the module and satellite species.

## Methods
### Population dynamics model
The GLV model represents the dynamics of $N$ interacting species as a set of ordinary differential equations,

$$\frac{dx_i}{dt} = r_i x_i + \sum_{j=1}^{N} a_{ij} x_i x_j, \, i = 1, \ldots, N. \quad (1)$$

Here, $x_i$ and $r_i$ are the abundance and intrinsic growth rate of species $i$, respectively, $a_{ij}$ quantifies how much species $i$ is affected by the abundance of species $j$, and $a_{ii} x_i^2$ represents the logistic growth term, which is set to $a_{ii} = -1$ for all $i$. We consider a microbial 'sample' as a steady state of a GLV model parameterized by the growth rate vector $\mathbf{r} = \{r_i\} \in \mathbb{R}^N$ and the interaction matrix $A = (a_{ij}) \in \mathbb{R}^{N \times N}$. Unless otherwise mentioned, in our simulations we set $N = 100$. The values of $\mathbf{r}$ are chosen from a uniform distribution $\mathbb{U}(0,1)$. The non-zero elements of $A$ define the 'interaction network' and are chosen to represent either Erdős-Rényi (ER)[42] or Barabási-Albert (BA)[43] topologies, as detailed below. The values of the non-zero elements of $A$ are drawn from a uniform distribution $\mathbb{U}(-\sigma_A, \sigma_A)$, where $\sigma_A$ represents a scaling factor chosen to ensure ecological stability. For each GLV model (defined by specific $A$ and $\mathbf{r}$), we generate cohorts of $M = 100$ different 'samples' (alternative steady states) by choosing different random initial conditions, i.e., a set of initial present species and abundances. Specifically, each species is initially present in a sample with a probability of 0.8, and the values of the initial abundances are chosen from a uniform distribution $\mathbb{U}(0,1)$. Each sample is simulated by integrating the differential equations (Eq. (1)) using the `ode45` function in MATLAB. The simulated samples are normalized to 1, i.e., only the relative abundance is used in the analysis of keystone species.

### Simulating keystone species
In our simulations, we create two different types of keystone species (see Fig. 2). 'Interaction-strength-based keystones' are created in structurally homogeneous (ER) interaction networks by boosting the strength of the out-going interactions of selected species. Alternatively, 'Structure-based keystones', are naturally created in heterogeneous (BA) interaction networks where a few species have much more interactions compared with all other species. We explain them both in detail below.

**Interaction-strength-based keystones.** We start by creating an interaction matrix $A$ that represents an ER model with edges density $p_{ER}$, in which each species interacts with a characteristic number $p_{ER}(N-1)$ of other species. Then, we enhance the strength of the out-going interactions $a_{ji}$ of species $i$, by multiplying them with a positive constant $K^i$,

$$\tilde{a}_{ji} = K^i a_{ji}, j = 1, \ldots, N, \quad (2)$$

where $\tilde{a}_{ji}$ are the new interactions strengths after the boosting procedure. These new interactions are then used in Eq. (1) to simulate the dynamics. The value of $K^i$ can be chosen manually to introduce a designated keystone species or can be randomly sampled from a long-tailed distribution (e.g., log-normal distribution)[44], which generates a few highly influential species whilst most other species have low interaction strength.

**Structure-based keystones.** Following ref. 22, to create Structure-based keystones, we use the BA model which generates a network with heterogeneous degree distribution. The model generates a network of $N$ nodes, based on $n_0$ pre-selected seed nodes. The seed nodes form a fully connected network, and the remaining $N - n_0$ nodes are sequentially connected to the $n < n_0$ existing nodes in the network, with probability $p_i$ proportional to the degree $d_i$ of each node $i$. A directionality parameter $d$ is also added to the process of reconstructing scale-free networks for the numerical simulations, which partially negates the independence between the out-going and in-going interactions[22]. The scale-free degree distribution in the network naturally creates keystones, i.e., nodes with a comparably large number of interactions, and are usually part of the seed nodes (see Supplementary Fig. 3). MATLAB script used to generate directed BA networks is presented in Supplementary Information.

### Calculating presence-impact in perturbation experiments
The presence-impact of species $i$ in a local community (sample) $k$ is defined as follows. We denote the abundance profile of a particular sample $k$ with $S_k$, in which species $i$ can be initially present or absent, while after a perturbation where species $i$ was added/removed, the abundance profile is denoted as $S_k^*$. We label the pre-perturbation abundance profile of all the species except species $i$ as $S_k^i$, and the post-perturbation abundance profile, excluding species $i$, as $S_k^{i*}$. Both $S_k^i$ and $S_k^{i*}$ are re-normalized to 1, to remove the mathematical relations between species $i$ and the rest of the species that are the result of the compositionality nature of the data, i.e.,

$$S_k^i(j) := \frac{S_k^i(j)}{\sum_j S_k^i(j)} \quad (3)$$

and similarly for $S_k^{i*}$. We then calculate the community-specific presence-impact, $I_k^i$, as the distance between $S_k^i$ and $S_k^{i*}$,

$$I_k^i = \langle S_k^i, S_k^{i*} \rangle, \quad (4)$$

where the $\langle \cdot \rangle$ symbol represents a distance function between the two samples, e.g., Bray–Curtis (BC) or root Jensen-Shannon divergence. The

distance $I_k^i$ represents the influence of the presence of species $i$ on the abundance profile of the rest of the species for the local community $k$.

The presence-impact of species $i$ is then the average of $I_k^i$ over a cohort of different communities. The presence-impact of species $i$, $I^i$, is defined as the average

$$I^i = \frac{1}{M} \sum_k I_k^i, \tag{5}$$

where $M$ is the number of samples. A high value of $I^i$ indicates that there is a large general difference between the abundance profiles associated with adding/removing species $i$. The presence-impact $I^i$ is a direct way of measuring the effect of removing/adding a species from the system which is suitable for both numerical and real-world experiments. Note that our definition is different from the one used in ref. 22. A MATLAB script for calculating $I$, in the case of GLV dynamics, is presented in the Supplementary Information.

### Identification of candidate keystones from cross-sectional data

Consider a cohort of $M$ samples, with abundance profiles $S_k$, $k = 1, ..., M$ which were either generated from GLV dynamics or are the result of real-world surveys. To calculate the EPI of species $i$, we divide the samples into two subsets, subset $G^i$ which includes all samples where species $i$ is present, and subset $\overline{G^i}$ which includes all samples where species $i$ is absent. Then, we test the level of separation between the two subsets based on the abundance profiles that include all species excluding species $i$. We have devised three alternative methods to measure this separation (see Fig. 1).

**Distance-based separation, $D_1$.** The separation between the two subsets, $G^i$ and $\overline{G^i}$, is measured as the average distance between the samples of the two subsets, $D_1^i$. We denote the abundance profiles in which species $i$ is present, excluding the abundance of species $i$ itself, as $S_k^i, k \in G^i$, and where species $i$ is absent as $S_{k'}^i, k' \in \overline{G^i}$. Both $S_k^i$ and $S_{k'}^i$ are re-normalized to 1 to minimize compositionality effects. Then, we calculate the average distance between $S_k^i$ and $S_{k'}^i$, namely $D_1^i$, as

$$D_1^i = \frac{1}{|G^i||\overline{G^i}|} \sum_{k \in G^i, k' \in \overline{G^i}} \langle S_k^i, S_{k'}^i \rangle, \tag{6}$$

where the sum is over all the possible pairs of samples from the $G^i$ and $\overline{G^i}$ subsets, and the $|\cdot|$ symbol denotes the size of the subset.

**Distance-based separation, $D_2$.** Alternatively, we can calculate the distance between the mean samples, $D_2^i$,

$$D_2^i = \left\langle \frac{1}{|G^i|} \sum_{k \in G^i} S_k^i, \frac{1}{|\overline{G^i}|} \sum_{k' \in \overline{G^i}} S_{k'}^i \right\rangle. \tag{7}$$

In Fig. 3a–f, we present the measured EPI values, $D_1$ and $D_2$, on GLV dynamics for a designated, interaction-strength-based keystone compared with a random, non-keystone species, showing a high level of separation between the subsets defined by the presence-absence of the keystone species. In cases where the impact of a species results in a uniform change of the microbiome composition, the outcomes of the two measures may coincide, as demonstrated for GLV dynamics in Fig. 3. The $D_1$ measure is suitable to the case where the samples with and without the keystone species are clustered into two distinct groups (in the space embedded by the re-normalized abundances of all other species), with large inter-cluster distance compared to the mean intra-cluster distances. The $D_2$ measure is more suitable to cases where the two groups of samples are not necessarily linearly separable, meaning that even if the two clusters are centered at the same point, the average distance between samples of different groups is large.

**Modularity-based separation, $Q$.** The distance-based approach mentioned above has a potential flaw. It assumes that the relations between all the different samples-pairs can be reliably measured using the same dissimilarity scale. This is not generally true for high-dimensional and non-linear spaces, and is exacerbated when calculating a dissimilarity value between 'distant' points.

An alternative approach would be to focus on short distances only and construct a sample-to-sample similarity network-based on the most similar sample pairs. The separation level between two subsets will then be measured based on their structural span over this network. To do this, we calculate the modularity, a measure of the correlation between the labels of the nodes in a network, and the structure of the network[45–47]. If the modularity is maximal, i.e., equals 1, then each node is only connected to other nodes with the same label. Low modularity indicates a high degree of mixing between the nodes, i.e., each node has a similar probability of being connected to a node with the same label as to a node with a different label. The modularity is calculated based on the structure of the sample-to-sample similarity network, independently from the similarity values calculated between very different samples, and is therefore more suited for detecting communities even when the embedded space is unusual or complex.

To apply the modularity method for species $i$, we first define a network of inter-samples similarities with respect to the abundances of all other species excluding species $i$, whereas the presence/absence of species $i$ defines the labels of the nodes. Specifically, we calculate the distances between the abundance profiles of the rest of the species for all the sample pairs $\langle S_\alpha^i, S_\beta^i \rangle$ where $\alpha, \beta \in \{1, ...M\}$ and $\alpha \neq \beta$. The nodes of the network represent the abundance profiles $S_\alpha^i$ and edges represent sample pairs with distance smaller than a threshold $T$ such that only a certain percentile $p_Q$ of the samples are connected. All abundance profiles, $S_\alpha^i$, are re-normalized to 1 to minimize compositionality effects.

The network associated with species $i$ is represented by an adjacency matrix $B^i$, where each element of the matrix $b_{\alpha\beta}^i$ is equal to 1 if $\langle S_\alpha^i, S_\beta^i \rangle \leq T$, and 0 otherwise. Then, each node $\alpha$ is labeled using a membership variable $s_\alpha^i$. If species $i$ is present in sample $\alpha$ (i.e., $\alpha \in \underline{G^i}$), then $s_\alpha^i = 1$. Otherwise, if species $i$ is absent from sample $\alpha$ (i.e., $\alpha \in \overline{G^i}$), then $s_\alpha^i = -1$. Finally, the modularity, $Q^i$, of the network associated with the presence/absence of species $i$, is given by

$$Q^i = \frac{1}{2w^i} \sum_{\alpha\beta} \left[ b_{\alpha\beta}^i - \frac{d_\alpha^i d_\beta^i}{2w^i} \right] \frac{s_\alpha^i s_\beta^i + 1}{2}, \tag{8}$$

where $d_\alpha^i$ is the degree of sample $\alpha$, $w^i$ is the total number of edges, $w^i = \frac{1}{2} \sum_\alpha d_\alpha^i$, and the sum is over all sample pairs. The value of $Q^i$ is bounded between −0.5 and 1, with larger values indicating a higher level of separation between the subsets[46]. Note that the modularity measured here is not the same as presented in ref. 48. There, the authors calculated the modularity of the interaction network (where each node represents an individual species). Here, the modularity is calculated in relation to the abundance profiles (each node represents a different sample).

A MATLAB script for calculating the EPI values $D_1^i$, $D_2^i$, and $Q^i$ of a given cohort of abundance profiles are presented in the Supplementary Information.

### Longitudinal EPI

The HMP data provides a unique opportunity to verify the results of the EPI, whilst eliminating issues of confounding factors, emanating from the possible heterogeneity of the different subjects. The process is based on comparing samples from the same subject, at two different time points. We denote the sample of subject $h$ of the first collection as $S_{h,\mathrm{I}}$ and of the second collection as $S_{h,\mathrm{II}}$. For each species $i = 1...N$, we check if the presence state of species $i$ is different between the first and

second collections, i.e., if species $i$ is present in the first collection and absent in the second collection, or vice versa. If the present state is indeed reversed, we define $S_{h,\mathrm{I}}^{i}$ and $S_{h,\mathrm{II}}^{i}$ as the abundance profiles of subject $h$ at the first and second collection times, respectively, excluding species $i$ and re-normalized to 1, i.e.,

$$\sum_{j=1,j\neq i}^{N} S_{h,\mathrm{I}}^{i}(j) = S_{h,\mathrm{II}}^{i}(j) = 1. \tag{9}$$

Then, for species $i$, and for subject $h$, the longitudinal EPI $L_{h}^{i}$ is the distance between the re-normalized first and second collections,

$$L_{h}^{i} = \langle S_{h,\mathrm{I}}, S_{h,\mathrm{II}} \rangle. \tag{10}$$

The process is then repeated for all $h = 1 \ldots H$ subjects to get the total longitudinal EPI

$$L^{i} = \frac{1}{H} \sum_{h=1}^{H} L_{h}^{i}. \tag{11}$$

We avoid low prevalent species with a high bias towards the longitudinal EPI by filtering out from the analysis species for which the measure was based on 10 or fewer subjects, i.e., at least in 11 subjects the presence state of the species was reversed between the first and second collections.

### Analysis of perturbation experiments

The process of identifying keystones through the use of presence-impact must include a simple, but necessary, statistical step. Analogously to the use of community importance from ref. 9, a species is said to be a keystone when its presence-impact is significantly large compared with the rest of the species. Therefore, when calculating the presence-impact for a given experiment, we also need to calculate the presence-impact of all other species individually and compare them to each other. We then consider a species to be a keystone species if its impact is larger than two standard deviations from the mean impact of all species. We demonstrate it in Fig. 2 where the impact of an artificial strength-based keystone, and structure-based keystone, are indeed much larger than their peers (blue and black arrows in Fig. 2b, d, respectively).

### Analysis of cross-sectional data

Similarly, a species is said to be a candidate keystone when its EPI value is significantly larger than the rest of the species. Indeed for simulated keystones, the high EPI value of the designated keystones is apparent for all three proposed measures (Fig. 3) compared with the values of the non-keystone species.

Unlike the ideal perturbation experiments, special considerations must be taken into account when calculating the EPI of species from high-throughput sequencing, mainly due to certain biases that stem from differences in the relative frequency of the species (i.e., the percent of samples where the species is present). For example, by definition, it is impossible to calculate the EPI of a species that is present in all the samples of a cohort, since we can not compare them to the case where it is absent (in these cases we expect to only be able to calculate the empirical abundance-to-community impact, which is beyond the scope of this work). Furthermore, the EPI may be affected by the frequency of the species of interest, a bias that is mainly apparent when it is either present or absent in only a small fraction of the samples. Supplementary Fig. 1 demonstrates this bias on a test case of simulated cross-sectional data with no inter-species interactions. To avoid this issue, when dealing with real-world data, we limited the analysis only to species with a sufficient number of present and absent samples (see "Human microbiome data filtering and analysis").

### Human microbiome data filtering and analysis

We analyzed real high-throughput sequencing of the human gut from the HMP[24,25]. Samples represent the abundances of operational taxonomic units (OTUs), obtained from 16S rRNA sequencing (variable regions V3 to V5), of $M = 107$ healthy human subjects. We consider individual OTUs as "species". The samples went through the following filtering process. We first filtered out any OTUs not present in any of the samples. We then ordered the OTUs according to their mean abundance and kept only the top $N = 1000$ OTUs with the top mean abundance. The abundances of the remaining OTUs in each profile were normalized to one. Finally, the presence-impact was calculated only for OTUs with a relative frequency between 0.25 and 0.75, which ensures maximal statistical power and avoids extreme cases with very high or very low frequencies that are prone to high bias in $D_1$ and $D_2$ measures (see Supplementary Fig. 1).

### Reporting summary

Further information on research design is available in the Nature Portfolio Reporting Summary linked to this article.

## Data availability

The Human Microbiome Project data used in this study are available in the HMP database at https://hmpdacc.org/. The *Arabidopsis* datasets were graciously provided by the research group of Dr. Vorholt. They are also available at the following references[29,30]. No original experimental data were collected in this study.

## Code availability

MATLAB scripts used in this study are available in the Supplementary Information and at: https://github.com/guy531/keystone[49].

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

## Acknowledgements

Our sincere gratitude goes out to Dr. Ruben Garrido-Oter and Dr. Christopher M. Field for their help in providing the data for Supplementary Fig. 8. We also thank Yang-Yu Liu, Jonathan Friedman, and Nadav Shnerb for their valuable contributions and insights. Lastly, we appreciate the useful comments of Boaz Amit, Dana Ben Porath, and Tal Ben Porath. A.B. thanks the Azrieli Foundation for supporting this research. This research was supported by the Israel Science Foundation (grant no. 1258/21) and the German-Israeli Foundation for Scientific Research and Development (grant no. I-1523-500.15/2021).

## Author contributions

G.A. and A.B. conceived the project and wrote the manuscript. G.A. performed the analysis. A.B. guided the project.

## Competing interests

The authors declare no competing interests.
