## [Peer Review File · Nature Communications]

Top-down identification of keystone taxa in the microbiomeReviewers' comments:

Reviewer #1 (Remarks to the Author):

This manuscript introduces a method to detect keystone species from metagenomic data of microbial communities. The authors introduce a new metric (or a new set of metrics), named empirical presence-impact (EPI), which quantifies the "effect" of the presence of a species on the abundance of other species. The value of EPI allow to correctly identify keystone species in simulated samples, where communities are separated in PCoA space thanks to the presence or absence of the identified keystone species. These methods are then applied to empirical data. In cross-sectional samples, the authors observe a separation in PCoA space based on the presence/absence of the keystone species. The impact measured in longitudinal data is significantly correlated with the values of the EPI measured in cross-sectional data.

The paper is well written and easy to follow. The methodology is well explained. I have some major concerns about the interpretation of the metric and the results.

1. I find the use of wording like "impact" (in EPI), and the interpretation of the value of EPI, which is potentially confusing. High values of EPI are interpreted as if the presence/absence of a species "caused" the variability in abundance of other species. This distinction between "cause" (presence/absence of a keystone) and "effect" (change in abundance profiles) cannot be inferred from observational data, in the same way as correlation does not imply causation. High values of EPI could be in fact also interpreted the other way: species with high EPI are the ones most affected by changes in abundance profiles. More generally, it is impossible to disentangle causal links from correlation in observational data. Since the definition of keystone species involve a causal effect and an identification through perturbation experiments, it appears to me that it is not possible to detect them from observational data.

2. It is unclear to me whether the separation of communities in PCoA space observed in Fig 4 b and e is surprising or a trivial consequence of the definition of EPI. In my understanding, EPI metrics are defined as the amount of variance/dissimilarity (measured in alternative ways) explained by the presence/absence of putative keystone species. These definitions are similar in spirit to how PCoA/PCA axes are identified.

3. It is unclear to me how to interpret the correlation between longitudinal and cross-sectional. The correlation is significant but quite weak. I understand that this weak correlation suggests that the EPI is not fully determined by differences across hosts. On the other hand, also the composition within hosts varies considerably over time. The "within host" variability could be an important factor in play, and, if this within-host variability has similar statistical features to the variability across host, one would expect a correlation between longitudinal impact and EPI.

Reviewer #2 (Remarks to the Author):

The authors present three new measures of "empirical presence impact" (EPI) to quantify "keystoneness" in microbial community data. EPI works well in simulations with the generalized Lotka-Volterra. However, the authors do not provide an evaluation on biological data with known keystones. Thus, given the shortcomings of the gLV (which the authors acknowledge), it is not clear whether these measures will work for real-world data.

There are very few data sets with experimentally validated keystone species. However, the few that exist (e.g. Paine's original work, <https://journals.plos.org/plosbiology/article?id=10.1371/journal.pbio.1002352>, <https://pubmed.ncbi.nlm.nih.gov/31558832/>,

<https://www.embopress.org/doi/full/10.15252/msb.20178157>) have not been used by the authors to validate their measures. The HMP data are not sufficient for validation, since we do not know yet which species are keystones in the human gut. It is an interesting application case, but does not replace a validation on biological data with known keystone species.

The gLV has a number of shortcomings, as described by the authors. One specific drawback of the gLV is that it cannot take into account how microorganisms will change their metabolism in response to a changed environment caused by species removal or addition. So why not using a more realistic, metabolite-explicit model that is better suited to this particularity of microbial communities, e.g. the stable marriage model (<https://www.ncbi.nlm.nih.gov/pmc/articles/PMC6246551/>), to check the new keystone measures? In general, it is better to test new methods on more than one microbial community model so as not to be biased by the assumptions of a single model.

Although they acknowledge that presence/absence status derived from sequencing data is problematic, the authors still rely on it. However, we know that sequencing depth will alter the presence/absence status. The authors should have carried out simulations with noisy data to look at the impact of wrongly assigned presence/absence status on the performance of the keystone identification measures.

The dependency of presence/absence status on sequencing depth is one of the reasons why relative-abundance-based measures are generally preferred in microbial sequencing data analysis - they do not depend as sensitively on sequencing depth. The problem of compositional data also affects presence/absence measures in the sense that only a limited number of reads can be sequenced, so whether a read of a rare species is sequenced will still depend on the abundance of other species.

The authors compute mutual information on presence/absence profiles to identify a keystone module. Does that keystone module also show up with a different, abundance-based (dis)similarity or correlation measure or is it specific to mutual information?

Reviewer #3 (Remarks to the Author):

In this manuscript, the authors proposed a mathematical framework to detect keystone species in the microbiome. While I like the overall idea of this work, I have several concerns that preclude my recommendation in its present form.

Major comments

Abstract: I am confused with the authors' assertion that the proposed top-down identification framework identifies keystone taxa without the assumption of pairwise interactions or any underlying dynamics. Isn't that an important role of keystone taxa? The influence of keystones on an ecological community would rely on their interactions with subsidiary taxa and/or such influence will also affect the interactions among subsidiary taxa. It gets more confusing as the authors claim that network response to perturbation can be assessed without understanding the underlying dynamics. The authors use directionality (parameter d) to calculate structural keystones. I think directionality inherently assumes underlying dynamics in the dataset. More importantly, directionality information is often not available in microbiome datasets, so the structural keystone framework may not be relevant to most microbiome studies.

Empirical Presence Impact in gut microbiome data: when applied to an actual microbiome dataset, the output seems fairly weak with little or no difference between candidate keystone and random species (Figure 4). Especially for D1 and Q, there is hardly any difference. I think it would be important to assess the robustness of this framework and test it on other microbiome datasets, preferably from complex environments.

Discussion: This section is very short and there is hardly any discussion. For example, the authors

can discuss how the proposed framework can be useful to formulate hypotheses in keystone taxa research. I would also like to see some discussions on the implications of this work for microbial network analysis of complex environmental systems. It would be nice to see discussions on the usefulness of this framework in terms of context dependency.

Non-linear interactions: While linear correlation networks are predominant in microbial ecology, non-linear interactions might be more common in the microbial world. It would be useful if the authors could include a few lines on this and how non-linearity could be included in the proposed framework.

The reviewer does not have the adequate mathematical background to assess the details in calculating the community-specific presence impact. However, it is unclear how the authors could separate presence from abundance to assess their impacts, especially, how to discern the presence-impact from abundance-impact.

Results-Page 5: Yes, species relative abundance in compositional data certainly has drawbacks, but so do presence/absence data. Given the issues associated with sequencing (e.g., MiSeq V3) and subsequent bioinformatic processing (OTU vs ESV), interpretations from presence/absence data can also be challenging.

On Page 12, the authors state that modularity is independent of spatial relationships between nodes. I have a naïve question about the environmental relevance of this assumption. Is modularity truly independent of spatial relationships? I think spatial proximity can enhance modularity in microbial networks. Please comment.

Page 5: "but have no classification at the species level"- this is the reason why keystone taxa should be used instead of keystone species when considering microbiome datasets. I suggest modifying the title accordingly.

Minor comments

It is difficult to comment on specific points without line numbers.

Please check the MS for grammatical and typographical errors. On P4, an extinction threshold; underlying dynamic structure; Page 9: the simulated 'samples' are normalized to? Throughout the MS-'than' the should be used and not 'then' the.

Structure-based keystones would be better than structural-based keystones.

NCOMMS-21-33784

“Top-down identification of keystone taxa in the microbiome” by Guy Amit and Amir Bashan

Point-by-point response to the Reviewers' comments

Response to Reviewer #1

This manuscript introduces a method to detect keystone species from metagenomic data of microbial communities. The authors introduce a new metric (or a new set of metrics), named empirical presence-impact (EPI), which quantifies the "effect" of the presence of a species on the abundance of other species. The value of EPI allow to correctly identify keystone species in simulated samples, where communities are separated in PCoA space thanks to the presence or absence of the identified keystone species. These methods are then applied to empirical data. In cross-sectional samples, the authors observe a separation in PCoA space based on the presence/absence of the keystone species. The impact measured in longitudinal data is significantly correlated with the values of the EPI measured in cross-sectional data.

The paper is well written and easy to follow. The methodology is well explained. I have some major concerns about the interpretation of the metric and the results.

We thank the reviewer for his/her positive assessment of our research and its novelty and we are pleased to hear that the reviewer appreciates the quality of the writing of the manuscript.

1. I find the use of wording like "impact" (in EPI), and the interpretation of the value of EPI, which is potentially confusing. High values of EPI are interpreted as if the presence/absence of a species "caused" the variability in abundance of other species. This distinction between "cause" (presence/absence of a keystone) and "effect" (change in abundance profiles) cannot be inferred from observational data, in the same way as

correlation does not imply causation. High values of EPI could be in fact also interpreted the other way: species with high EPI are the ones most affected by changes in abundance profiles. More generally, it is impossible to disentangle causal links from correlation in observational data. Since the definition of keystone species involve a causal effect and an identification through perturbation experiments, it appears to me that it is not possible to detect them from observational data.

The Reviewer raises a very important point and we completely agree with their comment. It is of course correct that it is impossible to disentangle causal relations from correlations in observational data.

In fact, this is a distinction that is often lacking in many papers that deal with network-based keystone species detection, where typically species with high centrality in observational-built correlation networks are automatically designated as *keystone species*. In contrast, in our manuscript, we attempt to make the distinction between causal effects and correlation more clear by defining high EPI species in observational data as **keystone candidates**, whilst species that have been identified as truly affecting the system in perturbation experiments as simply **keystones**.

Following the reviewer's comment, since the word "Impact" in the definition of the EPI can be confusing, we have opted to change the name of our measure to Empirical Presence-abundance Interrelation (EPI).

A paragraph was added to the results section of the revised manuscript to make this important distinction more explicit (Page 4 of the revised manuscript):

"This empirical association, which we term Empirical Presence-abundance Interrelation (EPI), can be detected from cross-sectional data to identify candidate keystone species... Note the distinction here between candidate keystones and actual keystone taxa. Since the EPI is measured only on cross-sectional data, it can not distinguish between correlation and causation effects. In other words, a taxon can have a large EPI value if it affects the other taxa disproportionately, or if it is affected disproportionately by the other species.

To disentangle this correlation from causation effects, perturbation experiments are required, where the actual presence-impact can be directly measured. The EPI proposes which species of interest will be leading candidates for further perturbation experiments.”

2. It is unclear to me whether the separation of communities in PCoA space observed in Fig 4 b and e is surprising or a trivial consequence of the definition of EPI. In my understanding, EPI metrics are defined as the amount of variance/dissimilarity (measured in alternative ways) explained by the presence/absence of putative keystone species. These definitions are similar in spirit to how PCoA/PCA axes are identified.

With this comment, the reviewer tackles the core reason for our method. We agree with the Reviewer that the spirit of PCoA/PCA methods is somewhat similar to the EPI, and this is why we complement the EPI result with PCoA plots. However, there are important differences between the methods.

- 1) If a particular species has a high characteristic abundance compared with the rest of the species, the samples with and without said species will be separated in the PCoA space, even without any real biological effects on the rest of the species (See figure below, which is also Supplementary Figure 6 in the revised manuscript). In contrast, the EPI is calculated over the renormalized abundance profiles of the other species so this mathematical effect of compositional data is removed.
- 2) Indeed, PCoA/PCA methods can capture global effects that separate all the samples along a single axis. However, real microbial communities are characterized by complex, non-linear interactions that can be highly context-dependent, so the presence/absence effect of a taxon is mostly captured by methods that focus on local effects, i.e., comparison between similar

communities. (for further discussion about nonlinear interactions in the microbiome, see [H. Lin et al, *Nat. Comm.* **13**, 4946 (2022)]).

In the revised manuscript we devoted three paragraphs to the Discussion (Page 9 of the revised manuscript), as well as a new figure (Supplementary Figure 6), for a detailed discussion on the differences between the EPI measure and the PCA/PCoA methods. We thank the Reviewer for raising this important question, and we think that this comparison helps to clarify the novelty of our method.

Supplementary Figure 6: **Comparison between the effect of keystone species and high-abundance species on the EPI method and PCA.** **a**, Interaction network with a keystone species (species number 5) marked in red. The interactions of the species (red lines) were multiplied by a constant to make them larger than the average species. **b**, PCA plot of 100 samples generated with the interaction network in **a**. Samples where the keystone species is present are marked by the red color. The separation between the samples is evident. **c**, PCA plot of the samples, with species number 5 removed from the data. The separation is still evident as the effect of the species is still present, even without considering the abundance of the keystone itself. **d**, The modularity EPI value, Q , for the different species. Species number 5 is significantly larger. **e**, Same as **a**, but for a non-keystone species. Instead, species number 5 has only elevated abundance levels. **f**, Even though the effect of the species, in this case, is not

significantly higher than an average species, the PCA plot is divided into the two types of samples, as the abundance of the species itself is enough to alter the abundance of the rest of the species due to normalization effects. **g**, After removing species number 5, the samples are indistinguishable in the PCA space. **h**, The modularity value, Q , does not indicate a keystone species in this case.

“In some cases, the effect of a keystone species can be captured using dimensional reduction techniques, as demonstrated in Fig. 3. Yet, this approach has two significant drawbacks. First, when the samples represent relative abundances, large variations between the samples can be caused by the variance of a high-abundant species which does not necessarily interact with the other species either directly or indirectly (see Supplementary Fig. 6). In contrast, the EPI of a particular species is calculated over the re-normalized abundance profiles excluding that species (3), such that the spurious correlations due to the compositionality of the data are removed. Second, the aim of the PCA/PCoA methods is to preserve the global structure of the data and thus may miss the local effect of the presence/absence of a species [34]. By local effects, we mean that the presence/absence effect is measured between communities with similar compositions. Such local effects are conceptually analogous to perturbation experiments where a single species is introduced to a community to measure how it affects that same community.

Focusing on the local effects is important due to two characteristics of real microbial communities. The species-species interactions may be context-dependent, i.e., the same species may have different effects in different microbial communities [35-38]. In addition, microbial interactions are inherently non-linear [39,40], and in fact, it has been recently shown that straightforward correlation analysis of pairwise interactions from cross-sectional abundance information carries only limited information on the real underlying interactions [14,41]. The PCA/PCoA methods are not designed to account for context-dependent and non-linear interactions [34].

The EPI measure D_1 potentially circumvents this difficulty as it does not assume the samples are divided into two distinct groups, separated along a single dimension, but rather takes into account more complex effects, as schematically demonstrated in Fig. 1b,c.

The Q is designed to be even more sensitive to non-linear effects, as it compares only neighboring samples when evaluating the effect of the species, methodologically similar to the t-SNE method. When analyzing real-world data, it is important to utilize all the methods in our arsenal to ensure robust statistical conclusions, and the EPI measure helps us enforce the validity of keystone identification protocols.”

3. It is unclear to me how to interpret the correlation between longitudinal and cross-sectional. The correlation is significant but quite weak. I understand that this weak correlation suggests that the EPI is not fully determined by differences across hosts. On the other hand, also the composition within hosts varies considerably over time. The "within host" variability could be an important factor in play, and, if this within-host variability has similar statistical features to the variability across host, one would expect a correlation between longitudinal impact and EPI.

We thank the reviewer for raising both legitimate concerns.

Regarding the observed correlation between the cross-sectional EPI and the longitudinal EPI, we agree that a correlation of 0.38 suggests that the variability of the two measures is not completely explained by the biological keystone-ness of the species, but rather by other factors. This may be partially explained by an increased sensitivity to noise due to a limitation of our methodology of comparing the cross-sectional EPI and the longitudinal EPI which reduces its statistical power.

This limitation stems from the fact that the calculation of the longitudinal EPI for a particular microbial species is possible only using a subset of the population for which that species' presence attribute changed between the first and second measurements.

We also agree with the referee that, while the host-related environmental factors are assumed to be more stable for longitudinal data, it is possible that some confounding factors are related to both across-hosts and within-host variability, leading to spurious correlations between the two calculated measures. This effect can be completely ruled out only by direct perturbation experiments.

We added a paragraph in the main text of the revised manuscript (Page 8) that acknowledges these limitations of our methodology:

“The methodology of comparing the cross-sectional EPI and the longitudinal presence-impact has two limitations. First, the calculation of the longitudinal presence-impact for a particular microbial species is possible only using a subset of the population for which the presence attribute of that species has changed between the first and second measurements. Second, although the host-related environmental factors are assumed to be more stable for longitudinal data, it is possible that some confounding factors are related to both across-hosts and within-host variability, leading to spurious correlations between the two calculated measures. This effect can be completely ruled out only by direct perturbation experiments.”

We thank the Reviewer for their helpful and constructive comments and hope we addressed them in an adequate manner.

Response to Reviewer #2

The authors present three new measures of "empirical presence impact" (EPI) to quantify "keystoneness" in microbial community data. EPI works well in simulations with the generalized Lotka-Volterra. However, the authors do not provide an evaluation on biological data with known keystones. Thus, given the shortcomings of the gLV (which the authors acknowledge), it is not clear whether these measures will work for real-world data.

There are very few data sets with experimentally validated keystone species. However, the few that exist (e.g. Paine's original work, <https://journals.plos.org/plosbiology/article?id=10.1371/journal.pbio.1002352>, <https://pubmed.ncbi.nlm.nih.gov/31558832/>, <https://www.embopress.org/doi/full/10.15252/msb.20178157>) have not been used by the authors to validate their measures. The HMP data are not sufficient for validation, since we do not know yet which species are keystones in the human gut. It is an interesting application case, but does not replace a validation on biological data with known keystone species.

The Reviewer here justly requests validation of the EPI method in real-world data. This is a difficult task that required the analysis and comparison of two different types of datasets: perturbation experiments and cross-sectional data.

Our first task was to find suitable data for such an analysis. The data we need must satisfy several conditions in order to verify our method: 1) It must include cross-sectional samples with diverse species compositions that an EPI can be measured for the different species in it. 2) It must include perturbation experiments that measure the keystoneness of each species directly. 3) The two experimental datasets must also be performed and processed for the same ecological model ecosystem and under similar lab protocols in order for them to be comparable. The Reviewer suggested some resources, which were very helpful in our search. Eventually, after a lengthy

search process, we were narrowed down to only a single option to verify our method with real-world data.

We performed a comparative analysis between two independent studies, conducted by the same lab group of Prof. Julia A. Vorholt from ETH Zurich, on the microbiome of the same plant model, *Arabidopsis*. One study involved the collocation of cross-sectional samples from wild *Arabidopsis*, and the second involved perturbation experiments which measured the keystone-ness of the taxa. Both studies followed the same collection and processing protocols for the microbial data.

The first study is “Functional overlap of the *Arabidopsis* leaf and root microbiota”, *Nature* 2015, by Yang Bai et al. In this paper, the researchers collected large amounts of cross-sectional samples of the leaves of the plant *Arabidopsis* from different natural environments and identified bacterial isolates which they term *At-LSPHERE*. The second study is “Synthetic microbiota reveals priority effects and keystone strains in the *Arabidopsis* phyllosphere”, *Nature Ecology and Evolution* 2019, by Carlström et al. In this study the research group of Prof. Vorholt revisited the microbial isolates of the leaves of *Arabidopsis*, and detected keystones with perturbation experiments of single-strain drop-outs.

Thanks to the help of Prof. Vorholt and her group members, Dr. Ruben Garrido-Oter and Dr. Christopher M. Field, we were able to obtain the data from both experiments and compare them. We calculated the EPI values for 336 individual bacterial species, using the cross-sectional data, and compared them to the “Effect-size” values calculated for 25 selected species from the single-strain drop-out experiments. From these lists, we had 9 species with both EPI and “Effect-size” values (most of the perturbation experiments have been performed for species that are always, or almost always, present in the wild and for which EPI is not applicable). As shown below, the EPI values and the Effect-size values are correlation ($r = 0.7926$ with $p = 0.00367$). These results have been added to the revised supplementary information (Supplementary Figure 8).

These results validate the EPI as a measure for identifying keystone **candidates** that can be later tested for being **actual keystones** using perturbation experiments. Designated experiments for measuring the keystone-ness of a larger number of taxa are required to further test EPI. We report these new results in the revised manuscript (Page 7):

“As previously stated, direct validation of the EPI requires perturbation experiments on a large number of taxa. In Supplementary Figure 8 we show additional analysis of keystone detection in the leaves microbiome of the plant model Arabidopsis. The “effect size” from perturbation experiments for a small number of selected species of one dataset (see 29 for further details) is compared with the EPI values calculated from cross-sectional data of a different experiment 30. We find a significant correlation between the two values ($r = 0.7926$ with $p = 0.00367$), suggesting that natural variability across samples is shaped by ecological interactions that can be observed by controlled perturbation experiments.”

Supplementary Figure 8: **Effect size of perturbation experiments versus EPI values from cross-sectional data from two independent studies.** We compare the results of two independent studies of the microbiome of Arabidopsis leaf. In the first, Bai et al [38] collected microbial samples of the leaf of wild Arabidopsis and categorized the taxa into operational units they called At-LSPHERE. In the second study by Carlström, Vorholt, et al [39], the authors performed perturbation experiments on synthetic Arabidopsis microbial environments and quantified keystone species by their effect size. We have compared the effect size from the perturbation experiments with the EPI Q values of the cross-sectional samples to every shared taxon that was present in both data sets (n=9). The Pearson coefficient of this analysis is $r = 0.7926$ with $p = 0.00367$.

The gLV has a number of shortcomings, as described by the authors. One specific drawback of the gLV is that it cannot take into account how microorganisms will change their metabolism in response to a changed environment caused by species removal or addition. So why not using a more realistic, metabolite-explicit model that is better suited to this particularity of microbial communities, e.g. the stable marriage model (<https://www.ncbi.nlm.nih.gov/pmc/articles/PMC6246551/>), to check the new keystone measures? In general, it is better to test new methods on more than one microbial community model so as not to be biased by the assumptions of a single model.

The reviewer requests checking the EPI method on other, more realistic, dynamical simulations. The reviewer's suggestions of the Stable Marriage (SM) model are particularly well-suited. This model has a unique feature, compared to other models of population dynamics – It does not have an underlying interaction network. Instead, the interactions between the species are indirect, caused by conflicts between the preferences lists of the different species and the nutrients.

This makes traditional detection methods, which are based on reconstructing a network model of the system, misleading as the system is not characterized by pairwise interactions.

In Supplementary Figure 7 (shown below in this letter), we show how we detect keystone species in the SM model using the EPI measure. In this kind of model, it is difficult to directly create keystones *a priori*, since one cannot simply choose a species to have high centrality in an interaction network. One way we found of creating artificial keystones is by adopting a method suggested by Goyal et al (Ref. [23] in the revised manuscript) where the preference lists of the species are correlated. This increases the likelihood of conflicts between the species, which, in turn, significantly limits the number of possible steady states of the system. In this case, a few species will have a greater effect on resolving these conflicts if they are not present in the system, i.e., they have higher EPI values. As shown in Supplementary Figure 7, in the case of random preference lists, where the differences between the influence of different species are only due to random fluctuations, the distribution of EPI values is quite narrow. In marked contrast, in the case of strong correlations between the preference lists, the distribution of EPI values is considerably broad, where the EPI values of a few species are much larger compared with the majority of the species, an indication of candidate keystone species.

Supplementary Figure 7: **Detecting keystone species in the stable marriage model using the EPI method.** We model microbial samples using the stable marriage model as described in Ref. [23]. **a**, When the species and nutrients have random preference lists, the distribution of EPI values has a major peak with a few species with high EPI values due to random fluctuations. **b**, When the preference lists are correlated, the removal of a single species has a higher probability to resolve conflicts and therefore to cause a significant shift in the balance of the other species. Such a frustrated system will be detected as a large tail in the EPI distribution of the species. Here, the correlation factor $\alpha=0.025$ (as described in the Supplementary Information of [23]), corresponds to an average Spearman correlation value between the preference lists of 0.96.

These new results are reported on page 5 of the revised manuscript:

“In Supplementary Figure 7 we show how the D_1 EPI measure is used to identify keystone species in the Stable Marriage model of ecological dynamics [23]. In this model, the species abundance is determined by preference lists, with

no explicit underlying interaction network, so detecting species using a reconstruction of the correlation network makes little sense. When the preference lists are correlated, the system is frustrated and keystone species with higher than average EPI values are prominent.”

Although they acknowledge that presence/absence status derived from sequencing data is problematic, the authors still rely on it. However, we know that sequencing depth will alter the presence/absence status. The authors should have carried out simulations with noisy data to look at the impact of wrongly assigned presence/absence status on the performance of the keystone identification measures. The dependency of presence/absence status on sequencing depth is one of the reasons why relative-abundance-based measures are generally preferred in microbial sequencing data analysis - they do not depend as sensitively on sequencing depth. The problem of compositional data also affects presence/absence measures in the sense that only a limited number of reads can be sequenced, so whether a read of a rare species is sequenced will still depend on the abundance of other species.

We follow the Reviewer's advice and show in the revised SI an analysis of the stability of the EPI values for varying degrees of sequencing depth. Sequencing depth was simulated using a process of resampling from a given community, where the detection probability of each species was determined by its relative abundance in the community. This process was performed for both simulated data using GLV model with artificial interaction-based keystones, and for real metagenomic data of the gut microbiome from the HMP.

As shown in Supplementary Figure 9 (copied below), the ranking of the EPI values, which determines which taxa are candidate keystones, is quite stable for large values of sequencing depth. In particular, species with large EPI have a tendency to stay at the top of the list. Indeed, when the sequencing depth is too shallow, the ranking is less stable, even for the top-ranked EPI. We discuss this important point in the Results section (Page 7) of the revised manuscript:

“In Supplementary Figure 9 we show how the ranking of the taxa according to the EPI values of the top keystone candidate, is relatively stable to the choice of sequencing depth in both simulated and real data from the human gut.”

Importantly, another reason we analyze the presence-impact of microbial taxa is not because of the quality of the data (in fact, our method combines both presence/absence data of the species in question and the relative abundance data of all other species), but rather for other methodological reasons. We refer the Reviewer to our reply to Reviewer #3 pages 22-24 of this letter.

Supplementary Figure 9: **EPI, D₁ values as a function of sequencing depth for simulated and real data.** a, Simulated GLV dynamics. b, Gut microbiome from the HMP. Species with large EPI have a tendency to preserve their high ranking even when the sequencing depth is increased.

The authors compute mutual information on presence/absence profiles to identify a keystone module. Does that keystone module also show up with a different,

abundance-based (dis)similarity or correlation measure or is it specific to mutual information?

We choose to use the mutual information measure as it is designed to detect cases of non-linear relationships of presence-absence data. Following the question of the Reviewer, we have repeated the analysis using the Jaccard similarity measure. The new figure is presented in the revised Supplementary Information, and copied below. As shown in Supplementary Figure 10 d-f, the correlation values for the cases of D1 and Q using the Jaccard similarity are qualitatively similar to those obtained using mutual information measure, while those of D2 are slightly weaker in the case of the Jaccard similarity.

Supplementary Figure 10: **Keystone modules in the presence-absence co-occurrence network with the Jaccard similarity measure.** Similar to Figure 6 but with the Jaccard similarity measure instead of the Mutual Information measure.

We thank the Reviewer for their helpful and constructive comments and hope we addressed them in an adequate manner.

Response to Reviewer #3

In this manuscript, the authors proposed a mathematical framework to detect keystone species in the microbiome. While I like the overall idea of this work, I have several concerns that preclude my recommendation in its present form.

We thank the Reviewer for his general positive assessment of our manuscript and for his helpful comments.

Major comments

Abstract: I am confused with the authors' assertion that the proposed top-down identification framework identifies keystone taxa without the assumption of pairwise interactions or any underlying dynamics. Isn't that an important role of keystone taxa? The influence of keystones on an ecological community would rely on their interactions with subsidiary taxa and/or such influence will also affect the interactions among subsidiary taxa. It gets more confusing as the authors claim that network response to perturbation can be assessed without understanding the underlying dynamics.

We acknowledge the lack of clarity on this important point in the previous version of the manuscript. We agree with the Reviewer that the keystone attribute is a result of inter-species interactions or underlying dynamics. Our core argument is that, indeed, it is possible to detect keystone species **without detailed knowledge of the pairwise interactions**. This is due to several reasons. In real-world microbial communities, it is generally unknown if the real underlying dynamics are in the form of pairwise interactions only, or, alternatively, it could be that higher-order interactions play a role, where three or more species influence each other in a hypergraph manner. Furthermore, we do not know the underlying mathematical form of the interactions, so trying to deduce the parameters of an assumed model might be futile.

Our proposed method avoids these pitfalls. We do not assume any type of interaction structure of equations. Instead, we find the effect of a single species on the rest of the

species in a broad manner. Similarly, we claim that the response of the abundance of all other species is enough to estimate the importance of the species in the underlying dynamics, even without knowing what it is exactly.

We have modified the abstract accordingly to increase legibility.

*“Keystone taxa in ecological communities are native taxa that play an especially important role in the stability of their ecosystem and can also be potentially used as its main drivers. However, we still lack an effective framework for identifying these taxa from the available metagenomic data without the notoriously difficult step of reconstructing the detailed network of inter-specific interactions. In addition, while most microbial interaction models assume pair-wise relationships, it is yet unclear whether pair-wise interactions dominate the system, or whether higher-order interactions are relevant. Here we propose a top-down identification framework, which detects keystones by their total influence on the rest of the taxa. **Our method does not assume a priori knowledge of pairwise interactions or any specific underlying dynamics** and is appropriate to both perturbation experiments and metagenomic cross-sectional surveys. When applied to real metagenomic data of the human gastrointestinal microbiome, we detect a set of candidate keystones and find that they are often part of a keystone module -- multiple candidate keystone species with correlated occurrence. The keystone analysis of single-time-point cross-sectional data is also later verified by the evaluation of two-time-points longitudinal sampling. Our framework represents a necessary advancement towards the reliable identification of these key players of complex, real-world microbial communities.”*

In addition, we further clarify this in the revised version of the Introduction (Page 3 of the revised manuscript):

“The effect of the species is measured without calculating the pair-wise correlation network, and does not even assume that the ecological dynamics are governed by pair-wise interactions.”

The authors use directionality (parameter d) to calculate structural keystones. I think directionality inherently assumes underlying dynamics in the dataset. More importantly, directionality information is often not available in microbiome datasets, so the structural keystone framework may not be relevant to most microbiome studies.

We thank the Reviewer for raising this point which was insufficiently explained in the previous version of the manuscript. When we say directionality in the context of modeling microbial interactions in numerical simulations. It is related to the specific way of building the scale-free network, which we adopted from [D. Berry and S. Widder, *Frontiers in microbiology*, 2014] (Ref. 22 in the manuscript). It is referring to the symmetry between the in-going and out-going edge distribution. If $d=1$, each outgoing edge between two species has also a corresponding ingoing edge. If $d=0$, there is no correlation between the ingoing and outgoing edges of any species, i.e., they are chosen independently. The reviewer is correct that directionality information is often not available in microbiome datasets. In fact, in any of the analyses of our EPI method, we did not presume to know this information, in both simulation and real-data analysis. We have revised the Methods section to clarify this point (Page 12 of the revised manuscript).

“A directionality parameter d is also added to the process of reconstructing scale-free networks for the numerical simulations, which partially negates the independence between the out-going and in-going interactions [22].”

Empirical Presence Impact in gut microbiome data: when applied to an actual microbiome dataset, the output seems fairly weak with little or no difference between candidate keystone and random species (Figure 4). Especially for $D1$ and Q , there is hardly any difference. I think it would be important to assess the robustness of this

framework and test it on other microbiome datasets, preferably from complex environments.

We thank the Reviewer for raising these important points. First, the EPI measures should not be interpreted according to their absolute values but rather their relative values. This means that the EPI value of a specific taxon should be compared with the entire distribution of all other species in its community. One way to do it could have been to normalize the EPI values and consider only their z-scores (the distance from the mean divided by the standard deviation) of the EPI. Yet, for the sake of transparency, in our manuscript, we prefer to report the raw EPI values, instead of the normalized z-scores. Figure 4 graphically illustrates this effect by highlighting the range of EPI values with $z\text{-score} > 2$, which shows that both for the D1 and Q measures there are significant keystone candidates.

We better explain this point in the main text of the revised manuscript (Page 6), which now reads:

“The figure shows the existence of candidate keystones in real data from stool samples, as exemplified by EPI values larger than two standard deviations compared with the rest of the species (marked by the shaded grey area). Note that keystone candidates are identified only by the relative values of their EPI compared with the cohort of the species, not by the absolute value.”

Second, as suggested by the Reviewer, we have performed additional analyses on real microbiome data, to assess the robustness of the EPI method:

- 1) We repeated the same analysis shown in Figure 4 for 12 additional cohorts of microbial samples from different body sites. These results are reported in Supplementary Figures 12-24 and are discussed in the revised manuscript:

Systematic analyses of additional 12 cohorts of microbial samples from different body sites are shown in Supplementary Figures 12-24. In most cases, we see that the distribution of EPI values has a dominant mean peak with a small number of species with considerably higher EPI values. In addition, the presence/absence of species with the top EPI value is associated with the separation of the samples in the PCoA plots and the sample-to-sample networks. Exceptional is the vaginal microbiome (Supplementary Figures 22-24), which is characterized by obvious clusters, or state types [28] (as also seen in the PCoA analysis). In these cases, the presence/absence patterns of many species are associated with the clusters, so no specific species stand out in the EPI distributions. In the Discussion section, we further discuss the relation between the EPI method and dimension reduction techniques, such as PCA or PCoA.

- 2) We analyzed microbial communities from the plant model *Arabidopsis* and found that the EPI values calculated using cross-sectional data are correlated with the effect size (a measure that was used by the original authors of the manuscript dataset to assess the keystone-ness of a species), calculated from direct perturbation experiments. See our response to Reviewer #2.

All in all, these additional analyses of the EPI measures on real microbial data further support the effectiveness of the EPI method as a tool for detecting keystone candidates under diverse experimental scenarios.

Discussion: This section is very short and there is hardly any discussion. For example, the authors can discuss how the proposed framework can be useful to formulate hypotheses in keystone taxa research. I would also like to see some discussions on the implications of this work for microbial network analysis of complex environmental systems. It would be nice to see discussions on the usefulness of this framework in terms of context dependency.

We thank the reviewer for their important suggestion. The discussion has been heavily revised accordingly to include these important points.

Non-linear interactions: While linear correlation networks are predominant in microbial ecology, non-linear interactions might be more common in the microbial world. It would be useful if the authors could include a few lines on this and how non-linearity could be included in the proposed framework.

We thank the reviewer for this helpful comment. We have revised the Discussion section accordingly to emphasize the issue of nonlinearity in microbial research, and how it relates to our method:

“Focusing on the local effects is important due to two characteristics of real microbial communities.

The species-species interactions may be context-dependent, i.e., the same species may have different effects in different microbial communities [35-38]. In addition, microbial interactions are inherently non-linear [39,40], and in fact, it has been recently shown that straightforward correlation analysis of pairwise interactions from cross-sectional abundance information carries only limited information on the real underlying interactions [14,41].

The PCA/PCoA methods are not designed to account for context-dependent and non-linear interactions [34].

The EPI measures SD_1 potentially circumvents this difficulty as it does not assume the samples are divided into two distinct groups, separated along a single dimension, but rather takes into account more complex effects, as schematically demonstrated in Fig. 1b,c.

The Q is designed to be even more sensitive to non-linear effects, as it compares only neighboring samples when evaluating the effect of the species, methodologically similar to the t-SNE method. When analyzing real-world data, it is important to utilize all the methods in our arsenal to ensure robust statistical

conclusions, and the EPI measure helps us enforce the validity of keystone identification protocols.”

The reviewer does not have the adequate mathematical background to assess the details in calculating the community-specific presence impact. However, it is unclear how the authors could separate presence from abundance to assess their impacts, especially, how to discern the presence-impact from abundance-impact.

The reviewer raises an important point. In our method, we examine how the microbial community **abundance** profile is affected by the **presence** of individual taxon. We do this by comparing samples with and without that individual. In this sense, the method combines both the abundance information and presence information of the taxa. The EPI, however, does not distinguish between high-abundance and low-abundance samples when calculating the impact. This is by design as it is a) much simpler to apply and avoid biases and 2) closer methodologically to perturbation experiments. It will be very interesting to have a *top-down* measure of the impact that takes into account abundance variation, and we plan to investigate it further in future research. We elaborate on the difference between presence-impact and abundance-impact in the revised version of the Discussion.

Results-Page 5: Yes, species relative abundance in compositional data certainly has drawbacks, but so do presence/absence data. Given the issues associated with sequencing (e.g., MiSeq V3) and subsequent bioinformatic processing (OTU vs ESV), interpretations from presence/presence data can also be challenging.

The Reviewer raises a key point. Indeed, presence/absence data is not necessarily superior to relative abundance data. It suffers from different drawbacks, as mentioned by the Reviewer. Note that for the calculation of the EPI we use a combination of both

the presence/absence data of the species in question and the relative abundance data of the other species.

Besides the advantages and drawbacks of presence/absence data, the presence state of a species in a microbial community is an important ecological feature of the system. We choose to focus on the presence/absence attribute of the species in question, mainly due to two methodological reasons: i) The original and commonly used ecological definition of keystone species relates to its presence/absence. ii) Currently available methods for manipulating microbial communities include tools that can introduce new taxa (probiotics and FMT) or remove taxa (narrow spectrum antibiotics), while directly controlling the abundance of specific species is currently not feasible.

In addition, since the sequencing depth is a major contributor to biases in presence/absence data [26], we tested how much the EPI ranking is affected by different values of sequencing depth in both simulations and real data. We found that the ranking of the taxa according to the EPI values of the top keystone candidate is relatively stable to the choice of sequencing depth in both simulated and real data from the human gut.

(Supplementary Figure 9, also discussed above in our response to Reviewer #2, pages 14-15 in this letter).

Finally, in the Discussion section and Supplementary Figure 11 we discuss the conceptual differences between presence-impact and abundance-impact and mention that future development of an effective measure for abundance-impact will supplement the analysis of keystone taxa.

To better explain the considerations and limitations of focusing on the presence/absence attribute, we heavily revised the first paragraph in the section “Presence-impact in real microbial communities” (Page 5 of the revised manuscript), which now reads:

“We analyze real metagenomic cross-sectional data of gastrointestinal tract from the Human Microbiome Project (HMP) [24, 25] (see Methods) and apply our EPI measures to identify keystone candidates. Estimation of species abundances through metagenomic sequencing surveys is susceptible to significant uncertainties due to many factors including experimental errors and sampling noise. This also affects the presence/absence pattern of the species. For example, it has been recently shown [26] that the species observed relative frequencies (the percent of samples where the species were detected) are determined by the mean and variance of their abundances, together with the sampling depth. Furthermore, the presence/absence of data is affected by the specific experimental and computational pipelines used. Thus, when analyzing real metagenomic data, the 'absence' of a species should be interpreted as being below the detection limit, whereas an observed species is more confidently defined as 'present'. Still, the presence/absence is commonly considered to be more robust than the estimated abundance [27]. In addition, it avoids the issue of comparing the species relative abundance in compositional data. The EPI measures use a combination of both the presence/absence of the species in question and the relative abundance data of the other species. We choose to focus on the presence/absence of the species in question, mainly due to two methodological reasons: i) The original and commonly used ecological definition of keystone species relates to its presence/absence. ii) Currently available methods for manipulating microbial communities include tools that can introduce new taxa (probiotics and FMT) or remove taxa (narrow spectrum antibiotics), while directly controlling the abundance of specific species is currently not feasible.”

On Page 12, the authors state that modularity is independent of spatial relationships between nodes. I have a naïve question about the environmental relevance of this assumption. Is modularity truly independent of spatial relationships? I think spatial proximity can enhance modularity in microbial networks. Please comment.

We thank the Reviewer for pointing this out. By spatial relationship we mean spatial in the sample-space of the abundance of the species (the R^N space), not the 3-dimensional space the species are inhabiting in the real world. In this sense, the modularity is indeed independent of the spatiality as it measures the separability of the nodes in the N-dimensional space without taking into account the distances between the clusters. In the revised manuscript we have rephrased this sentence Methods section.

The modularity is calculated based on the structure of the sample-to-sample similarity network, independently from the similarity values calculated between very different samples, and is therefore more suited for detecting communities even when the embedded space is unusual or complex.

Page 5: "but have no classification at the species level "- this is the reason why keystone taxa should be used instead of keystone species when considering microbiome datasets. I suggest modifying the title accordingly.

We thank the reviewer for their constructive suggestion. The manuscript and the title have been revised accordingly.

Minor comments

It is difficult to comment on specific points without line numbers.

Line numbers have been added to the revised manuscript for the convenience of the reader.

Please check the MS for grammatical and typographical errors. On P4, an extinction threshold; underlying dynamic structure;

We thank the reviewer. Comprehensive editing has been applied to correct grammatical errors.

Page 9: the simulated 'samples' are normalized to?

Normalized to 1, i.e., the sum of the abundance of all the species of any sample is 1. The manuscript has been revised accordingly.

Throughout the MS-'than' the should be used and not 'then' the.

We thank the Reviewer for their careful reading. The manuscript has been revised accordingly.

Structure-based keystones would be better than structural-based keystones.

We agree with the Reviewer's suggestion and have revised the manuscript accordingly.

We thank the Reviewer for their helpful and constructive comments and hope we addressed them in an adequate manner.

REVIEWER COMMENTS

Reviewer #2 (Remarks to the Author):

The authors have adequately addressed a number of my previous comments. However, I still have a serious concern about their method, as described below.

Major issue

Environmental factors as main drivers behind differential species abundances are potentially a main confounder of the EPI method. Species with high EPI values may rather be indicators than keystone species. I suggest the authors check how strong a confounder the environment can be by introducing species-specific environmental effects in their gLV simulations.

L. 246-248 "A reasonable hypothesis is that one or few of the species in the module is indeed a keystone species, and the rest are 'satellite species', which themselves do not have an unusually large presence-impact, but are strongly connected to the keystone species."

Another reasonable hypothesis is that the species respond to environmental change, and the species in the same module share a common response, without any keystone species being involved. This is the reason why it is helpful to include metadata in network construction (such as host properties, dietary components if available, Bristol stool score, etc.). Also, to what extent do keystone modules depend on the network inference method being used?

Minor issues

What is the rationale to compute both D1 and D2? What is the interpretation in case they differ?

L. 43-44 "in contrast to presence-impact experiments, abundance-impact perturbation might be practically impossible in real-world experiments"

Why? Species addition is not that hard.

L. 162-163

"Still, the presence/absence is commonly considered to be more robust than the estimated abundance [27]. In addition, it avoids the issue of comparing the species relative abundance in compositional data."

I do not find this argument convincing. First, one reference is not sufficient to claim that something is "commonly considered". I disagree with the authors on the higher robustness of presence/absence vs relative abundance for network construction and hub node identification. Absence in sequencing data doesn't necessarily mean absence from the sample. Also, a species that is present may appear to be absent because another species bloomed, so the problem of compositionality doesn't go away simply by looking at presence/absence. I think it's better to write that the opinions on that subject are divided or to drop the argument altogether. Only a proper evaluation will clarify that matter.

L. 168-169: "while directly controlling the abundance of specific species is currently not feasible"

Not in the sense of maintaining species at a selected abundance value, but species addition (when the species is already present) is a direct way to manipulate its abundance. Prebiotics also manipulate species abundance rather than presence/absence.

Code:

Matlab scripts hidden away in the supplement aren't very user-friendly. I ask the authors to provide an easy-to-access and install, easily usable, well-documented, and free tool together with a tutorial/study case illustrating its use. Since Matlab isn't freely available, the authors can consider a free Matlab alternative such as Octave (<https://octave-online.net/>) for this purpose or provide a proper R package.

Reviewer #3 (Remarks to the Author):

The revised manuscript is definitely an improved version. The authors have adequately addressed the questions/concerns on the previous version. The new figures are elegant, and the clarifications are convincing. I only have a pending comment about the presence-impact and abundance impact and a couple of other minor comments.

Presence impact vs abundance impact: the importance of keystone species should be irrespective of their actual abundance, and this is a key distinction from dominant species. There are well-known keystone pathogens in the human microbiome that are considered rare taxa, however, there are also examples of keystones that are highly abundant. However, it gets a bit confusing as the authors state “that we define the presence-impact through the abundance change of the other species.” in Ln 114. So, the presence impact is also based on the abundance to some extent? To test the importance of a selected keystone species, I suggest that authors plot how network properties and community importance change with increasing abundance of that keystone species. While I appreciate Supp Fig 11, I think it would be useful if the authors perform this on a real microbiome dataset. This will reveal if only their presence is adequate for their effect, or they have a specific abundance threshold to exert their influence.

Secondly, the authors state that “The EPI measures D_1 potentially circumvents this difficulty as it does not assume the samples are divided into two distinct groups, separated along a single dimension”. However, Figure 1C assumed exactly the same right?

I suggest using ‘high-throughput sequencing’ data instead of metagenomic data because the data used here are essentially amplicon sequencing data.

Response to Reviewers

In the provided letter, the Reviewers' comments are colored in **blue**, our responses are colored in **black**, and new additions to the manuscript are colored in **red**.

Response to Reviewer #2

The authors have adequately addressed a number of my previous comments. However, I still have a serious concern about their method, as described below.

Major issue

Environmental factors as main drivers behind differential species abundances are potentially a main confounder of the EPI method. Species with high EPI values may rather be indicators than keystone species. I suggest the authors check how strong a confounder the environment can be by introducing species-specific environmental effects in their gLV simulations.

L. 246-248 "A reasonable hypothesis is that one or few of the species in the module is indeed a keystone species, and the rest are 'satellite species', which themselves do not have an unusually large presence-impact, but are strongly connected to the keystone species."

Another reasonable hypothesis is that the species respond to environmental change, and the species in the same module share a common response, without any keystone species being involved. This is the reason why it is helpful to include metadata in network construction (such as host properties, dietary components if available, Bristol stool score, etc.). Also, to what extent do keystone modules depend on the network inference method being used?

We thank the Reviewer for these important comments. Indeed, environmental factors can play an important part in the identification of keystone species.

Following the Reviewer suggestion, we have carried out simulation that mimic the effects of environmental factors in the identification of keystone species. The new results and figure have been added to the Supplementary Information of the revised manuscript and are discussed in the Results section of the main manuscript (page 8).

We have performed new simulations where a subset of the species is affected by the environment, in a species-specific fashion. The environment influence on this subset of species can be positive or negative, affecting both their presence probability and their characteristic abundance. Samples are simulated assuming heterogeneous environmental conditions.

The following new parameters that determine the effects of the environment are added to the gLV model:

<code>num_affected_species</code>	The number of species that "sense" the environment, i.e., that the external environment influences their presence and abundance. An integer between 1 and N (the total number of species)
<code>sample_environment</code>	A local factor (unique for each sample) that determines the local environment of each

	sample. A binary number (0/1) that divides the samples into two types (environment A or environment B).
<code>enviromental_effect</code>	A global factor that represents the strength of the environmental effect in environment B .

Then, we perform the following simulation: We create $M = 90$ samples from gLV dynamics, each with $N = 100$ species. The interaction matrix is identical to all the samples. Additionally, a species-specific base growth rates, r_0 , are chosen from a uniform distribution between 0 and 1. We randomly choose `num_affected_species` number of species that are affected by the environment (The figure shows results for `num_affected_species = 1, 5, 10` and 50). Half of the samples are randomly chosen to be of type *A* and the rest are of type *B*. Type *A* environment has no effect on the growth rate or the presence/absence of any of the species. In contrast, type *B* environment does affect the growth rate and the presence/absence of the subset of species that were randomly chosen to be affected by the environment. The effect of the type *B* environment is to multiply the base growth rate of the affected species by the factor `enviromental_effect` (positive effect) or by the reciprocal of that parameter (negative effect), at random for each affected species. As a result, the characteristic abundances of some species are typically larger in environment *B* and some are smaller, compared with their abundances in environment *A*. Additionally, the presence probabilities of the affected species in environment *B* are different from environment *A*. In environment *A* all species have the same probability of being present, 50%. In environment *B* some of the affected species have a 80% change of being present, whilst others have a 30% chance.

In general, we find two types of keystone candidates in this scenario. ‘Actual’ keystones, which were randomly prominent in the interaction network, and ‘fake’ keystones, which had a large EPI value purely due to the environmental confounding factors. The environmental effect is however not trivial. For example, if only a single species is affected by the environment, it will not be registered as a keystone candidate (i.e., its EPI value will not be unusually large) since the abundance of the rest of the species will not be correlated with its presence/absence. When a moderate number of species are affected by the environment, then fake keystone candidates will be registered. That is because the presence/absence of the affected species are correlated with the abundance modulation of many other species (which are affected as well). Finally, when a very large proportion of the species is affected by the environment, then fake keystones will *not* be registered. That is since, although the presence/absence of species *are* correlated with abundance modulation of many other species, this effect will not be *unusually* large, and so the scheme of detecting keystone by comparison will fail to detect this impact. In other words – when too many species are ‘fake’ keystones, no species is fake keystone.

These simulations highlight two important facts: 1. The environment can absolutely be a confounding factor responsible for the false detection of keystone species. We therefore stress once more that until controlled perturbation experiments are performed, no species should ever be considered a ‘true’ keystone, but only a keystone *candidate*. 2. When the environmental effect is *too* broad, and when the keystone detection scheme only considers relative importance of species (as opposed to absolute importance), environmental factors can be mitigated in the detection of candidates.

We also agree with the Reviewer that where the environment affects the presence/absence patterns of several species in a similar way, they might be falsely detected as candidate keystones. Such shared presence/absence pattern can be represented as a module in a network of presence/absence similarities. We note that the networks we present do not aim to infer the actual species-species interrelations, a tedious task that requires comparison of different inference methods. Rather, these networks intend to demonstrate a *proof-of-concept* that keystone modules might be important in the keystone identification step, by representing a specific aspect of the species-species interrelations, i.e., the similarity of their presence-absence patterns.

We would like to take this opportunity to again thank the Reviewer for this important insight which greatly enriched the conclusions of our manuscript.

We have added the following figure to the Supplementary Information.

Supplementary Figure 25: An example of environmental confounding factors affecting keystone candidates' detection in GLV dynamics. Simulations that mimic the effects of environmental factors in the identification of keystone species. Each sample belongs to environment A or environment B. Both environments have the same underlying interaction network. Species that are colored in red are affected by the type of environment, meaning that their growth rate is randomly increased or decreased by a factor of 2 (top figures) or 50 (bottom figures). Also, their proclivity to be present in the samples is changed randomly from 50% in environment A to 30% or 80% in environment B. Species that are colored in blue are not affected by the environment. When only one species is affected by the environment (a and e), its likelihood to be detected as a keystone candidate is small, due to the fact that the abundance of all the other species is not spuriously correlated with its presence. Similarly, when many species are affected by the environment (d and h), their relative importance in the D_2 measure is reduced. When a few species are affected, then the spurious correlations are strong enough to be detected as keystone candidates, as indicated by their D_2 value being larger than two standard deviations from the mean (grey area).

Minor issues

 What is the rationale to compute both D_1 and D_2 ? What is the interpretation in case they differ?

We thank to the Reviewer for raising this point. The following text was added to the main text to clarify this:

In cases where the impact of a species results in a uniform change of the microbiome composition, the outcomes of the two measures may coincide, as demonstrated for GLV dynamics in Fig. 3. The D_1 measure is suitable to the case where the samples with and without the keystone species are clustered into two distinct groups (in the space embedded by the renormalized abundances of all other species), with large inter-cluster distance compared to the mean intra-cluster distances. The D_2 measure is more suitable to cases where the two groups of samples are not necessarily linearly separable, meaning that even if the two clusters are centered at the same point, the average distance between samples of different groups is large.

L. 43-44 "in contrast to presence-impact experiments, abundance-impact perturbation might be practically impossible in real-world experiments"

Why? Species addition is not that hard.

We appreciate the Reviewer comment. The difference between the plausibility of presence-impact and abundance-impact perturbations stems from their different ecological processes. The presence of a species in a specific environment might be due to ecological processes such as historical contingency, dispersal limitation or random sampling, while the abundance of that species is determined by a complex dynamic process that integrates the ecological interactions with the environment and the other species, leading to ecological balance. Therefore, while species addition (using probiotics), and even promoting the abundance of already present species (prebiotics), is indeed not difficult, controlling the exact amount of the abundance of the new species that will be introduced into the environment is practically very challenging, if not impossible. However, eliminating a species entirely, or introducing it into the environment without trying to control its exact abundance, is possible and is done regularly in both experimental and clinical settings. That is why we chose to focus on this more reasonable case in this paper. We hope that in the future we can produce other analysis and experimental methods that can reliably account for the abundance fluctuations in the discovery of keystone taxa.

L. 162-163

"Still, the presence/absence is commonly considered to be more robust than the estimated abundance [27]. In addition, it avoids the issue of comparing the species relative abundance in compositional data."

I do not find this argument convincing. First, one reference is not sufficient to claim that something is "commonly considered". I disagree with the authors on the higher robustness of presence/absence vs relative abundance for network construction and hub node identification. Absence in sequencing data doesn't necessarily mean absence from the sample. Also, a species that is present may appear to be absent because another species bloomed, so the problem of compositionality doesn't go away simply by looking at presence/absence. I think it's better to write that the opinions on that subject are divided or to drop the argument altogether. Only a proper evaluation will clarify that matter.

The Reviewer is correct that the argument is not very convincing, and that the problem of compositionality is not solved by considering only the presence/absence data. We have accepted the Reviewer suggestion to rewrite this section, and it now reads thus:

There are, however, arguments in favor of the robustness of presence/absence data in high-throughput sequencing [27], specifically with regards to the problem of compositionality, however the opinions on that subject are still divided.

L. 168-169: "while directly controlling the abundance of specific species is currently not feasible"

Not in the sense of maintaining species at a selected abundance value, but species addition (when the species is already present) is a direct way to manipulate its abundance. Prebiotics also manipulate species abundance rather than presence/absence.

We refer the Reviewer to our previous response to the comment on L. 43-44.

Code:

Matlab scripts hidden away in the supplement aren't very user-friendly. I ask the authors to provide an easy-to-access and install, easily usable, well-documented, and free tool together with a tutorial/study case

illustrating its use. Since Matlab isn't freely available, the authors can consider a free Matlab alternative such as Octave (<https://octave-online.net/>) for this purpose or provide a proper R package.

We thank the Reviewer for their suggestion. The code has been uploaded online to github. The “Code Availability” section has been revised to reflect that fact. The main functions are of a .m format which will work natively on Octave as well.

Response to Reviewer #3

The revised manuscript is definitely an improved version. The authors have adequately addressed the questions/concerns on the previous version. The new figures are elegant, and the clarifications are convincing. I only have a pending comment about the presence-impact and abundance impact and a couple of other minor comments.

We thank the Reviewer for acknowledging their appreciation with the improved version of the manuscript.

Presence impact vs abundance impact: the importance of keystone species should be irrespective of their actual abundance, and this is a key distinction from dominant species. There are well-known keystone pathogens in the human microbiome that are considered rare taxa, however, there are also examples of keystones that are highly abundant. However, it gets a bit confusing as the authors state “that we define the presence-impact through the abundance change of the other species.” in Ln 114. So, the presence impact is also based on the abundance to some extent? To test the importance of a selected keystone species, I suggest that authors plot how network properties and community importance change with increasing abundance of that keystone species. While I appreciate Supp Fig 11, I think it would be useful if the authors perform this on a real microbiome dataset. This will reveal if only their presence is adequate for their effect, or they have a specific abundance threshold to exert their influence.

We thank the Reviewer for raising this interesting point. Yes, the presence impact is based on the abundance to some extent, but not on the abundance of the analyzed species, but specifically on the renormalized abundance **of all the other species**. In this sense, abundance-impact will be a measure that tests how the abundance variations of a single species are related to the abundance profile of all the other species (an abundance-to-abundance relationship).

(The following text was added to the revised manuscript:)

To analyze the relation between the presence-impact of a species and its abundance in real microbiome datasets, we have plotted the D_2 values versus the mean abundance of all species in all the datasets we analyzed from the HMP repository. The following figure shows the results (Supplementary Figure 26 in the revised manuscript) as well as the associated Pearson correlation coefficients.

Supplementary Figure 26 demonstrates rich results. In some body sites, the species' EPI value seems to be correlated with their abundances (Left antecubital fossa and right antecubital fossa, panels g and l). On other body sites, while the overall correlation is low, the top abundant species are also among those with high EPI values (for example in the stool samples, panels a and c). Yet, in most cases, the EPI value seems to be independent upon the abundance, and the species with the highest EPI values, i.e., the keystone candidates, are not among the most abundant species.

Supplementary Figure 26: D_2 versus mean abundance for all HMP datasets. a, Stool sample - second visit b, Plaque c, stool samples - first visit d, Anterior nares e, Buccal mucosa f, Hard palate g, Left Antecubital fossa h, Left Retroauricular crease i, Mid vagina j, Palatine tonsils k, Posterior fornix l, Right Antecubital fossa m, Right Retroauricular crease n, Saliva o, Subgingival plaque p, Supragingival plaque q, Throat r, Tongue dorsum, s, Vaginal introitus. At the top of the figure the Pearson values r is shown together with the p value. While a significant correlation does exist for some data sets, it is clear that high presence impact cannot be explained fully by simply having a large abundance.

Secondly, the authors state that “The EPI measures D_1 potentially circumvents this difficulty as it does not assume the samples are divided into two distinct groups, separated along a single dimension”. However, Figure 1C assumed exactly the same right?

We agree with the Reviewer, however, Figures 1B and 1C are meant to show a simplified sketch of how the measure works. In this case we did choose to show in both figures the same illustration of two distinct groups for the sake of simplicity and to focus on the difference between the D_1 and D_2 measures.

I suggest using ‘high-throughput sequencing’ data instead of metagenomic data because the data used here are essentially amplicon sequencing data.

We thank the Reviewer for the suggestion. We have replaced metagenomic data with high-throughput sequencing in all the relevant locations.

REVIEWERS' COMMENTS

Reviewer #2 (Remarks to the Author):

The authors have addressed all the issues I raised. In my opinion, the manuscript is ready for publication.

Reviewer #3 (Remarks to the Author):

The authors have addressed my concerns, and do not have any further comments.